# Learning Discrete Latent Variable Structures with Tensor Rank Conditions

**Zhengming Chen**[1,2]**, Ruichu Cai**[1,*]**, Feng Xie**[3]**, Jie Qiao**[1]**,**
**Anpeng Wu**[2,4]**, Zijian Li**[2]**, Zhifeng Hao**[1,5]**, Kun Zhang**[2,6,*]

1. School of Computer Science, Guangdong University of Technology
2. Machine Learning Department, Mohamed bin Zayed University of Artificial Intelligence
3. Department of Applied Statistics, Beijing Technology and Business University
4. Department of Computer Science and Technology, Zhejiang University
5. College of Science, Shantou University, Shantou, Guangdong, China
6. Department of Philosophy, Carnegie Mellon University

## Abstract

Unobserved discrete data are ubiquitous in many scientific disciplines, and how to learn the causal structure of these latent variables is crucial for uncovering data patterns. Most studies focus on the linear latent variable model or impose strict constraints on latent structures, which fail to address cases in discrete data involving non-linear relationships or complex latent structures. To achieve this, we explore a tensor rank condition on contingency tables for an observed variable set $\mathbf{X}_p$, showing that the rank is determined by the minimum support of a specific conditional set (not necessary in $\mathbf{X}_p$) that d-separates all variables in $\mathbf{X}_p$. By this, one can locate the latent variable through probing the rank on different observed variables set, and further identify the latent causal structure under some structure assumptions. We present the corresponding identification algorithm and conduct simulated experiments to verify the effectiveness of our method. Our results elegantly expand the application scope of causal discovery with latent variables.

## 1 Introduction

Social scientists, psychologists, and researchers from various disciplines are often interested in understanding causal relationships between the latent variables that cannot be measured directly, such as depression, coping, and stress (Silva et al., 2006). A common approach to grasp these latent concepts is to construct a measurement model. For instance, experts design a set of measurable items or survey questions that serve as indicators of the latent variable and then use them to infer causal relationships among latent variables (Bollen, 2002; Bartholomew et al., 2011; Cui et al., 2018).

Numerous approaches exist for addressing structure learning among latent variables. In particular, if the data generation process is assumed to be a linear relationship, known as *linear latent variable models*, several approaches have been developed. These include the second-order statistic-based approaches (Silva et al., 2006; Kummerfeld and Ramsey, 2016; Chen et al., 2024; Sullivant et al., 2010), high-order moments-based ones (Xie et al., 2020; Chen et al., 2022; Cai et al., 2019; Adams et al., 2021), matrix decomposition-based methods (Anandkumar et al., 2013, 2014, 2015), and copula model-based approaches (Cui et al., 2018). Moreover, the hierarchical latent variable structure has been well-studied within the linear setting (Huang et al., 2022; Xie et al., 2022; Chen et al., 2023; Jin et al., 2023). However, the linear assumption is rather restrictive and the discrete data in the real world could be more frequently encountered (e.g., responses from psychological and educational

---

[*]Corresponding author

38th Conference on Neural Information Processing Systems (NeurIPS 2024).

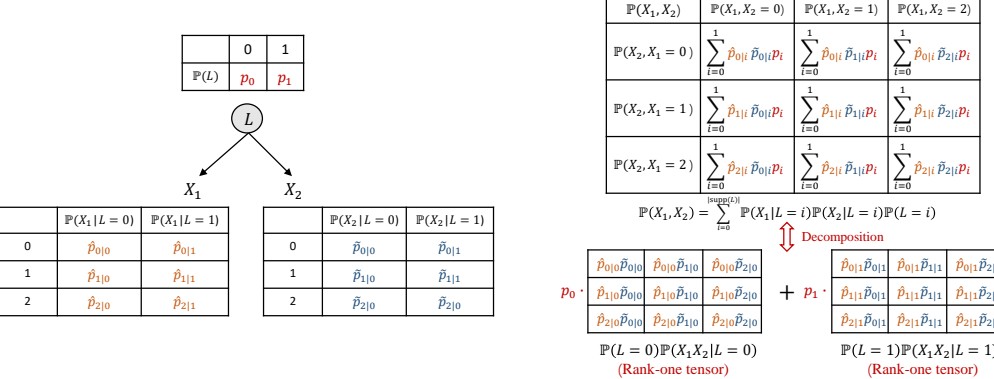

(a) A latent structure with conditional probability tables.  (b) The decomposition of the joint distribution.

Figure 1: Illustrating the graphical criteria of the tensor rank condition, the rank of the joint distribution is determined by the support of a specific conditional set that d-separates all observed variables, i.e., $\mathrm{Rank}(\mathbb{P}(X_1, X_2)) = |\mathrm{supp}(L)| = 2$. See Example 3.4 for details.

assessments or social science surveys (Eysenck et al., 2021; Skinner, 2019)), which does not satisfy the linear assumption.

When the data generation process is discrete, however, due to the challenging nonlinear transition relationship in discrete data, few identifiability results exist and are mostly only applicable in strict cases. In particular, under some prespecified structure, the identifiability of parameters is established, such as in the hidden Markov model(HMM) (Anandkumar et al., 2012) model, topic models (Anandkumar et al., 2014), and multiview mixtures model (Anandkumar et al., 2015). By further specifying the latent variable structure as a tree, Wang et al. (2017); Song et al. (2013) show that the structural model is identifiable. Recently, Gu (2022); Gu and Dunson (2023) further considered the identifiability of pyramid structure under the condition that each latent variable has at least three observed children. However, challenges persist in extending identifiability to more general structures among discrete latent variables. Existing approaches, unfortunately, cannot identify the causal structure of latent variables as shown in Fig. 2(a).

Recently, some studies have shown that causal structures involving discrete latent confounders can be effectively identified, building on the identifiability results of mixture models, as discussed in (Kant et al., 2024; Gordon et al., 2023; Mazaheri et al., 2023; Anandkumar et al., 2012). Most of these works focus on the causal structure among observed variables, usually assuming a single latent confounder. For the identification of latent structure, (Kivva et al., 2021) shows that causal structures of discrete latent variables can be identified by recovering the latent distribution from a mixture oracle. However, while a general discrete latent variable model can be identified theoretically, estimating the parameters of the mixture model is challenging. Approximating methods are often applied, but these may be unrealistic and impractical in real-world situations.

In this paper, we aim to establish a general identification criterion for discrete latent structures in cases where latent structures exhibit flexible dependencies, while also developing a simple but robust structure learning algorithm. To achieve this, we explore a tensor rank condition on the contingency tables for an observed variable set $\mathbf{X}_p$, to probe the latent causal structure from observed data. Interestingly, as shown in Fig. 1, we found that the rank of the contingency tables of the joint distribution $\mathbb{P}(X_1, X_2)$ is deeply connected to the support of a variable $L$ (not necessary among $X_1, X_2$) that d-separate $X_1$ and $X_2$. By this observation, we first develop a general tensor rank condition for the discrete causal model and show that such a rank is determined by the minimal support of a specific conditional set (not necessary in $\mathbf{X}_p$) that d-separates all variables in $\mathbf{X}_p$. Such findings intrigue the possibility to identify the discrete latent variables structure. We further propose a discrete latent structure model that accommodates more general latent structures and shows that the discrete latent variable structure can be identified locally and iteratively through tensor rank conditions. Subsequently, we present an identification algorithm to complete the identifiability of discrete latent structure models, including the measurement model and the structure model. We theoretically show that under proper causal assumptions, such as faithfulness and the Markov assumption, the

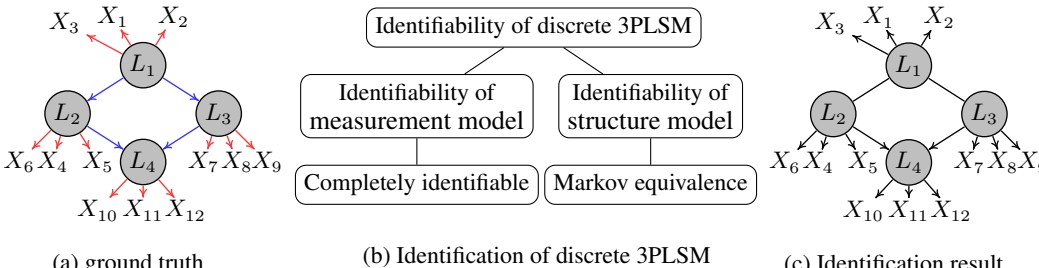

(a) ground truth    (b) Identification of discrete 3PLSM    (c) Identification result

Figure 2: An example of discrete latent structure model involving 4 latent variables and 12 observed variables (sub-fig (a)). Here, the red edges form a measurement model, while the blue edges form a structural model. The theoretical result of this paper is shown in sub-fig (c).

measurement model is fully identifiable and the structure model can be identified up to a Markov equivalence class.

The contributions of this work are three-fold. (1) We first establish a connection between the tensor rank condition and the graphical patterns in a general discrete causal model, including specific d-separation relations. (2) We then exploit the tensor rank condition to learn the discrete latent variable model, allowing flexible relations between latent variables. (3) We present a structure learning algorithm using tensor rank conditions and demonstrate the effectiveness of the proposed algorithm through simulation studies.

## 2 Discrete Latent Structure Model

For an integer $m$, denote $[m] = \{1, 2, \cdots, m\}$. Consider a discrete statistic model with $k$ latent variable set $\mathbf{L} = \{L_1, \cdots, L_k\}, L_i \in [r_i]$ and $m$ discrete observed variable set $\mathbf{X} = \{X_1, \cdots, X_m\}$ with $X_i \in [d_i]$ $(r_i, d_i \geq 2)$, in which any marginal probabilities are non-zero. We say a discrete statistic model is a *discrete causal model* if and only if $\mathbf{V} = \mathbf{L} \cup \mathbf{X}$ can be represented by a directed acyclic graph (DAG), denoted by $\mathcal{G}$. We use $\mathrm{supp}(V_i) = \{v \in \mathbb{Z}^+ : \mathbb{P}(V_i = v) > 0\}$ to denote the set of possible values of the random variable $V_i$. Our work is in the framework of causal graphical models. Concepts used here without explicit definition, such as d-separation, which can refer to standard literature (Spirtes et al., 2000).

In this paper, we focus on learning causal structure among latent variables in one class of discrete causal models. The model is defined as follows.

**Definition 2.1** (Discrete Latent Structure Model with Three-Pure Children). *A discrete causal model is the Discrete Latent Structure Model with Three-Pure Children (Discrete 3PLSM) if it further satisfies the following three assumptions:*

*1) [Purity Assumption] there is no direct edges between the observed variables;*

*2) [Three-Pure Child Variable Assumption] each latent variable has at least three pure variables as children;*

*3) [Sufficient Observation Assumption] The dimension of observed variables support is larger than the dimension of any latent variables support.*

These structural constraints inherent in the discrete 3PLSM are also widely used in linear latent variable models, e.g., Silva et al. (2006); Kummerfeld and Ramsey (2016); Cai et al. (2019); Xie et al. (2020). In the binary latent variable case, recently, a similar definition is also employed in Gu and Dunson (2023); Gu (2022). The key difference is that there are no constraints on the latent structure in our work. An example of a discrete 3PLSM model is shown in Fig. 2(a), where $L_1, \cdots, L_4$ represent discrete latent variables, and $X_1, \cdots, X_{12}$ are discrete observed (measured) variables.

Generally speaking, the discrete 3PLSM model can be divided into two sub-models (Spirtes et al., 2000), i.e., the measurement model and the structure model, e.g., red edge and blue edge in Fig. 2 (a). By this, one can first identify the measurement model to determine the latent variables and then use the measured variable to infer the causal structure of latent variables. As shown in Fig. 2 (b), we will separately discuss the identification of the two sub-models and show that the measurement model is

fully identifiable and the structure model is identified up to a Markov equivalence class. The symbols used in our work is summarised in Table 1.

To ensure the identification of causal structure and the asymptotic correctness of identification algorithms, some common causal assumptions are required.

**Assumption 2.2** (Causal Markov Assumption). *Let $\mathcal{G}$ be a causal graph with vertex set $\mathbf{V}$ and $\mathbb{P}_{\mathbf{V}}$ be probability distribution over the vertices in $\mathbf{V}$ generated by $\mathcal{G}$. We say $\mathcal{G}$ and $\mathbb{P}_{\mathbf{V}}$ satisfy the Causal Markov Assumption if and only if for every $V_i \in \mathbf{V}$, $\mathbb{P}(V_i, \mathbf{V} \setminus \mathrm{Des}_{V_i} | \mathrm{Pa}_{V_i} = i) = \mathbb{P}(W | \mathrm{Pa}_{V_i} = i)\mathbb{P}(\mathbf{V} \setminus \mathrm{Des}_{V_i} | \mathrm{Pa}_{V_i} = i)$.*

**Assumption 2.3** (Faithfulness Assumption). *Let $\mathcal{G}$ be a causal graph with vertex set $\mathbf{V}$ and $\mathbb{P}_{\mathbf{V}}$ be probability distribution over the vertices in $\mathbf{V}$ generated by $\mathcal{G}$. We say $< \mathcal{G}, \mathbb{P}_{\mathbf{V}} >$ satisfies the Faithfulness Assumption if and only if (i). every conditional independence relation true in $\mathbb{P}_{\mathbf{V}}$ is entailed by the Causal Markov Assumption applied to $\mathcal{G}$, and (ii). for any joint distribution $\mathbb{P}(\mathbf{L}_p)$, there does not exist $\mathbb{P}(\mathbf{L}_q)$ with $|\mathrm{supp}(\mathbf{L}_q)| < |\mathrm{supp}(\mathbf{L}_p)|$ such that $\mathbb{P}(\mathbf{L}_p) = \mathbb{P}(\mathbf{L}_q)$.*

**Assumption 2.4** (Full Rank Assumption). *For any conditional probability $\mathbb{P}(X | \mathrm{Pa}_X)$, the corresponding contingency table is full rank, i.e., each column of $\mathbb{P}(X | \mathrm{Pa}_X)$ is linearly independent with the other column vectors in the parameter space.*

In general, Assumptions 2.2 ~ 2.3 are widely used in the constraint-based causal discovery methods, e.g., PC algorithm and FCI algorithm (Spirtes et al., 2000; Spirtes and Glymour, 1991). One can see that we further constraint the parameter space of joint distribution cannot be reduced to a low-dimension space, for maintaining the diversity of parameter space. This is also the reason for Assumption 2.4, which aligns with the non-degeneracy condition used in (Kivva et al., 2021).

**Our goal:** The goal of our work is to develop a robust approach for identifying discrete latent structure models, including both the measurement and structural models.

| | |
|---|---|
| $\mathbf{V}$: The set of variables $\mathbf{V} = \mathbf{X} \cup \mathbf{L}$ | $\mathbf{X}$: The set of observed variables |
| $\mathbf{L}$: The set of latent variables | $V_i \perp\!\!\!\perp V_j | \mathbf{V}_p$ : Conditional independence |
| $|\mathbf{X}_p|$ : Dimension of $\mathbf{X}_p$ | $\mathcal{T}_{(\mathbf{X}_p)}$ : the tensor form of $\mathbb{P}(\mathbf{X}_p)$ |
| $\mathbb{P}(\mathbf{X}_p)$: the joint distribution of $\mathbf{X}_p$ | $\mathrm{Rank}(\mathcal{T}_{(\mathbf{X}_p)})$ : The rank of tensor $\mathcal{T}_{(\mathbf{X}_p)}$ |
| $\mathrm{Pa}_X$ : The parent set of $X$ | $\mathrm{Des}_X$ : The descendant set of $X$ |
| $\mathrm{Diag}(\mathrm{M})$: The diagonal matrix of M | $\mathbf{u}_i \otimes \mathbf{u}_j$ : The outer product of two vectors |

Table 1: Mathematical notations used in this paper.

# 3 Tensor Rank Condition with Graphical Criteria

To address the identification problem in the discrete 3PLSM, this section introduces the building block–the tensor rank condition of the discrete causal model. Then, we establish the connection between tensor rank and d-separation relations under a general discrete causal model.

Before formalizing the tensor rank condition, we first give the explicit definition of tensor rank.

**Definition 3.1** (Rank-one Tensor). *An $n$-way tensor $\mathcal{T} \in \mathbb{R}^{I_1 \times \cdots \times I_n}$ is a rank-one tensor if it can be written as the outer product of $n$ vectors, i.e.,*

$$\mathcal{T} = \mathbf{u}_1 \otimes \mathbf{u}_2 \otimes \cdots \otimes \mathbf{u}_n,$$

*where $\mathbf{u}_i$ are vectors that each represent a dimension of the tensor, $\otimes$ represents the outer product.*

**Definition 3.2** (Tensor Rank Kolda and Bader (2009)). *For an $n$-way tensor $\mathcal{T} \in \mathbb{R}^{I_1 \times \cdots \times I_n}$, the rank of a tensor $\mathcal{T}$ is defined as the **smallest** number of rank-one tensors that sum to exactly represent $\mathcal{T}$. Formally, the rank of tensor $\mathcal{T}$, denoted $\mathrm{rank}(\mathcal{T})$, is the smallest integer $r$ such that:*

$$\mathcal{T} = \sum_{i=1}^{r} \mathbf{u}_1^{(i)} \otimes \mathbf{u}_2^{(i)} \otimes \cdots \otimes \mathbf{u}_n^{(i)},$$

*where each $\mathbf{u}_k^{(i)}$ is a vector in the corresponding vector space associated with the $k$-th mode of $\mathcal{T}$.*

In other words, the tensor rank denotes the minimal number of rank-one decompositions. In the discrete causal model, the joint distribution can be represented as a tensor, e.g., the joint distribution of two random variables is a two-way contingency tensor. Interestingly, by carefully analyzing the (non-negative) rank-one decomposition of the joint distribution, we find that the tensor rank essentially reveals structural information within the causal graph. The result is presented below.

**Theorem 3.3** (Graphical implication of tensor rank condition). *In the discrete causal model, suppose Assumptions 2.2 ~ Assumption 2.4 hold. Consider an observed variable set $\mathbf{X}_p = \{X_1, \cdots, X_n\}$ ($\mathbf{X}_p \subseteq \mathbf{X}$ and $n \geq 2$) and the corresponding $n$-way probability tensor $\mathcal{T}_{(\mathbf{X}_p)}$ that is the tabular representation of the joint probability mass function $\mathbb{P}(X_1, \cdots, X_n)$. Then, $\mathrm{Rank}(\mathcal{T}_{(\mathbf{X}_p)}) = r$ ($r > 1$) if and only if (i) there exist a conditional set $\mathbf{S} \subset \mathbf{V}$ with $|\mathrm{supp}(\mathbf{S})| = r$ that d-separates any pair of variables in $\{X_1, \cdots, X_n\}$, and (ii) does not exist conditional set $\tilde{\mathbf{S}}$ that satisfies $|\mathrm{supp}(\tilde{\mathbf{S}})| < r$.*

We further provide an example to illustrate Theorem 3.3.

**Example 3.4** (Illustrating the graphical criteria). *Consider a single latent variable structure as shown in Fig. 1 (a) where $L$ is a latent variable with $\mathrm{supp}(L) = \{0, 1\}$ and $X_1, X_2$ are observed variables with $\mathrm{supp}(X_i) = \{0, 1, 2\}, i \in \{1, 2\}$. For convenience, we denote $p_i = \mathbb{P}(L = i)$, $\hat{p}_{i|j} = \mathbb{P}(X_1 = i|L = j)$, and $\tilde{p}_{i|j} = \mathbb{P}(X_2 = i|L_1 = j)$. The joint distribution $\mathbb{P}(X_1, X_2)$ can be expressed as the product of conditional probabilities, as shown in Fig. 1(b). By applying the tensor decomposition, we observe that $\mathbb{P}(X_1, X_2)$ can be decomposed as the sum of two rank-one tensors: $\mathbb{P}(X_1, X_2|L = 0)$ and $\mathbb{P}(X_1, X_2|L = 1)$. Thus, the rank of the tensor $\mathbb{P}(X_1, X_2)$ is two, corresponding to the cardinality of the latent variable's support. The reason $\mathbb{P}(X_1, X_2|L = i)$ is a rank-one tensor is that $L$ d-separates $X_1$ and $X_2$, i.e., $\mathbb{P}(X_1, X_2|L = i) = \mathbb{P}(X_1|L = i) \otimes \mathbb{P}(X_2|L = i)$. This illustrates the connection between tensor rank and d-separation relations.*

Intuitively, the graphical criteria theorem suggests that, in the discrete causal model, the tensor rank condition implies the minimal conditional probability decomposition within the probability parameter space, which hopefully induces the structural identifiability of the discrete 3PLSM model.

# 4    Structure Learning of Discrete Latent Structure Model

In this section, we address the identification problem of the discrete 3PLSM model using a carefully designed algorithm that leverages the tensor rank condition. Specifically, we first show that latent variables can be identified by finding causal clusters among observed variables (Sec. 4.1). Then, we use these causal clusters to conduct conditional independence tests among latent variables based on the tensor rank condition, identifying the structure model (Sec. 4.2). Finally, we discuss the practical implementation of testing tensor rank (Sec. 4.3). For simplicity, we focus on the case where all latent variables have the same number of categories. The result can be directly extended to cases with different numbers of categories (see details in Appendix E).

## 4.1    Identification of the measurement model

To answer the identification of the measurement model, one common strategy is to find the causal cluster that shares the common latent parent, which has been well-studied within the linear model, such as Silva et al. (2006); Cai et al. (2019); Xie et al. (2020). We follow this strategy and show that, in the discrete 3PLSM, the causal cluster can be found by testing the tensor rank conditions iteratively. The definition of a causal cluster is as follows.

**Definition 4.1** (Causal cluster). *In the discrete 3PLSM, the variable set $\{X_1, \cdots, X_n\}$ is a causal cluster, termed $C_i$, if and only if all variables in $\{X_1, \cdots, X_n\}$ share the common latent parent.*

It is not hard to see that, the measurement model can be identified if all causal cluster is found. In order to find these causal clusters by making use of the tensor rank condition, the key issue is to determine the support of latent variables in advance. This issue can be addressed by identifying the rank of the two-way tensor formed by the joint distribution of two observed variables.

**Proposition 4.2** (Identification of support of latent variables). *In the discrete 3PLSM model suppose Assumptions 2.2 ~ Assumption 2.4 hold. The support of the latent variable corresponds to the rank of the two-way probability contingency table for any pair of observed variables $X_i$ and $X_j$, i.e., $|\mathrm{supp}(L)| = \mathrm{Rank}(\mathcal{T}_{(X_i, X_j)}), \forall X_i, X_j \in \mathbf{X}$.*

This result holds because any pair of variables in the discrete 3PLSM model is d-separated by any one of their latent parent variables, and all latent variables have the same support. Next, we formalize the property of clusters and give the criterion for finding clusters.

**Proposition 4.3** (Identification of causal cluster). *In the discrete 3PLSM mode, suppose Assumption 2.2 ~ Assumption 2.4 hold. Let $r = |\mathrm{supp}(L_i)|$ denote the cardinality of the latent support. Given three disjoint observed variables $X_i, X_j, X_k \in \mathbf{X}$,*

- *$\mathcal{R}ule1$: if the rank of tensor $\mathcal{T}_{(X_i,X_j,X_k)}$ is not equal to $r$, i.e., $\mathrm{Rank}(\mathcal{T}_{(X_i,X_j,X_k)}) \neq r$, then $X_i$, $X_j$ and $X_k$ belong to the different latent parents.*
- *$\mathcal{R}ule2$: for any $X_s$, $X_s \in \mathbf{X} \setminus \{X_i, X_j, X_k\}$, if the rank of tensor $\mathcal{T}_{(X_i,X_j,X_k,X_s)}$ is $r$, i.e., $\mathrm{Rank}(\mathcal{T}_{(X_i,X_j,X_k,X_s)}) = r$, then $\{X_i, X_j, X_k\}$ share the same latent parent.*

**Example 4.4** (Finding causal clusters). *Let's take Fig. 2(a) as an example. One can find that for $\{X_1, X_2, X_3, X_k\}$, where $X_k \in \mathbf{X} \setminus \{X_1, X_2, X_3\}$, the rank of tensor $\mathcal{T}_{(X_1,X_2,X_3,X_k)}$ is $r$. Thus, $\{X_1, X_2, X_3\}$ is identified as a causal cluster.*

Next, we consider the practical issues involved in determining the number of latent variables by causal clusters. That is, there are some causal clusters that should be merged because they share one latent parent. We find that the overlapping clusters can be directly merged into one cluster. This is because the overlapping clusters have the same latent variable as the parent under the discrete 3PLSM model. The validity of the merge step is guaranteed by Proposition 4.5.

**Proposition 4.5** (Merging Rule). *In the discrete 3PLSM model, for two causal clusters $C_1$ and $C_2$, if $C_1 \cap C_2 \neq \varnothing$, then $C_1$ and $C_2$ share the same latent parent.*

Based on the above results, one can iteratively identify causal clusters and apply the merger rule to detect all latent variables. The identification procedure is summarized in Algorithm 1.

---

**Algorithm 1** Finding the causal cluster

---

**Input**: Data from a set of measured variables $\mathbf{X}_{\mathcal{G}}$, and the dimension of latent support $r$
**Output**: Causal cluster $\mathcal{C}$

1: Initialize the causal cluster set $\mathcal{C} := \varnothing$, and $\mathcal{G}' = \varnothing$;
2: *// Identify Causal Skeleton*
3: **Begin** the recursive procedure
4: **repeat**
5:     **for** each $X_i, X_j$ and $X_k \in \mathbf{X}$ **do**
6:         **if** $\mathrm{Rank}(\mathcal{T}_{\{X_i,X_j,X_k\}}) \neq r$ **then**
7:             **Continue**;   *// $\mathcal{R}ule1$ of Prop. 4.3*
8:         **end if**
9:         **if** $\mathrm{Rank}(\mathcal{T}_{\{X_i,X_j,X_k,X_s\}}) = r$, for all $X_s \in \mathbf{X} \setminus \{X_i, X_j, X_k\}$ **then**
10:            $\mathbf{C} = \mathbf{C} \cup \{\{X_i, X_j, X_k\}\}$;
11:         **end if**
12:     **end for**
13: **until** no causal cluster is found.
14: *// Merging cluster and introducing latent variables*
15: Merge all the overlapping sets in $\mathbf{C}$ by Prop. 4.5.
16: **for** each $C_i \in \mathbf{C}$ **do**
17:     Introduce a latent variable $L_i$ for $C_i$;
18:     $\mathcal{G} = \mathcal{G} \cup \{L_i \to X_j | X_j \in C_i\}$.
19: **end for**
20: **return** Graph $\mathcal{G}$ and causal cluster $\mathcal{C}$.

---

**Theorem 4.6** (Identification of the measurement model). *In the discrete 3PLSM, suppose Assumption 2.2 ~ Assumption 2.4 hold. The measurement model is fully identifiable by Algorithm 1.*

### 4.2 Identification of the structure model

Once the measurement model is identified, the observed children can serve as proxies for the latent variables, enabling the identification of the causal structure among them. Here, we employ constraint-based framework to learn the causal structure of latent variables.

Constraint-based structure learning algorithms find the Markov equivalence class over a set of variables by making decisions about independence and conditional independence among them. Given a pure and accurate measurement model with at least two measures per latent variable, we can test for independence and conditional independence (CI) among the latent variables. Specifically, to test statistical independence between discrete variables, one can examine whether the rank of their joint distribution contingency table is one (Sullivant, 2018). For testing conditional independence (CI) relations among latent variables, further leveraging the algebraic properties of the tensor rank condition is required (see Theorem 4.7).

**Theorem 4.7** (d-separation among latent variable). *In the discrete 3PLSM, suppose Assumption 2.2 ~ Assumption 2.4 hold. Let $r$ denote the cardinality of the latent support. Then, $L_i \perp\!\!\!\perp L_j | \mathbf{L}_p$ if and only if $\mathrm{Rank}(\mathcal{T}_{(X_i, X_j, \mathbf{x}_{p1}, \mathbf{x}_{p2})}) = r^{|\mathbf{L}_p|}$, where $X_i$ and $X_j$ are the pure children of $L_i$ and $L_j$, $\mathbf{X}_{p1}$ and $\mathbf{X}_{p2}$ are two disjoint child sets of $\mathbf{L}_p$ that satisfy $\forall L_i \in \mathbf{L}_p, \mathrm{Ch}_{L_i} \cap \mathbf{X}_{p1} \neq \varnothing, \mathrm{Ch}_{L_i} \cap \mathbf{X}_{p2} \neq \varnothing$.*

Intuitively, based on the graphical criteria of tensor rank condition, $\mathbf{L}_p$ is a minimal conditional set in the causal graph that d-separates $\mathbf{X}_{p1}$ and $\mathbf{X}_{p2}$ and hence the rank of tensor $\mathcal{T}_{(X_i, X_j, \mathbf{x}_{p1}, \mathbf{x}_{p2})}$ is the dimension of support of $\mathbf{L}_p$, if $X_i$ and $X_j$ also be d-separated by $\mathbf{L}_p$.

**Example 4.8** (CI test among latent variables). *Consider the structure in Fig. 2(a) and suppose $r = 2$. By selecting $\mathbf{X}_{p1} = \{X_4, X_7\}$ and $\mathbf{X}_{p2} = \{X_5, X_8\}$ as two disjoint child sets of $\{L_2, L_3\}$ respectively, let $\mathbf{X}_p = \{X_1, X_{10}, X_4, X_5, X_7, X_8\}$ and $\mathbf{L}_p = \{L_2, L_3\}$. One can see that the rank of tensor $\mathcal{T}_{(\mathbf{X}_p)}$ is four since $\mathbf{L}_p$ (i.e., $\{L_2, L_3\}$) is minimal conditional set that d-separates any pair variable in $\mathbf{X}_p$, which imply that $L_1 \perp\!\!\!\perp L_4 | \{L_2, L_3\}$.*

Based on Theorem 4.7, we introduce the PC-TENSOR-RANK algorithm. This method accepts a measurement model learned by the previous procedure, and outputs the Markov equivalence class of the structural model associated with the latent variables within the measurement model, in accordance with the PC algorithm. The implementation is summarised as Algorithm 2. Consequently, we establish the identification of the structure model as shown in Theorem 4.9.

---

**Algorithm 2** PC-TENSOR-RANK

---

**Input**: Data set $\mathbf{X} = \{X_1, \ldots, X_m\}$ and causal cluster $\mathcal{C}$
**Output**: A partial DAG $\mathcal{G}$.
1: Initialize the maximal conditions set dimension $k$;
2: Let $L_i$ denote as $C_i, C_i \in \mathcal{C}$;
3: Form the complete undirected graph $\mathcal{G}$ on the latent variable set $\mathbf{L}$;
4: **for** $\forall L_i, L_j \in \mathbf{L}$ and adjacent in $\mathcal{G}$ **do**
5:    *//Test the CI relations among latent variables by Theorem 4.7*
6:    **if** $\exists \mathbf{L}_p \subseteq \mathbf{L} \setminus \{L_i, L_j\}$ and $(|\mathbf{L}_p| < k)$ such that $L_i \perp\!\!\!\perp L_j | \mathbf{L}_p$ hold **then**
7:       delete edge $L_i - L_j$ from $G$;
8:    **end if**
9: **end for**
10: Search V structures and apply meek rules Meek (1995).
11: **return** a partial DAG $\mathcal{G}$ of latent variables.

---

**Theorem 4.9** (Identification of structure model). *In the discrete 3PLSM, suppose Assumption 2.2 ~ Assumption 2.4 hold. Given the measurement model, the causal structure over the latent variable is identified up to a Markov equivalent class by the PC-TENSOR-RANK algorithm.*

### 4.3 Practical Test for Tensor Rank

In our theoretical results, the key issue is to test the rank of a tensor, which involves estimating the dimension of latent support and the rank of a tensor. Here, we aim to explore methods to (i) estimate the rank of the contingency matrix for determining the dimension of latent support, and (ii) apply the goodness-of-fit test to assess the tensor rank.

**Estimate the rank of contingency matrix.** We start with the estimation of the dimension of latent variables support based on Prop. 4.2. There are many practical approaches used to estimate the rank of a general matrix M, such as Camba-Méndez and Kapetanios (2009). In our implementation, we

Table 2: Results on learning pure measurement models, where the data is generated by the discrete 3PLSM. Lower value means higher accuracy.

| Algorithm | | Latent omission | | | | Latent commission | | | | Mismeasurements | | | |
|---|---|---|---|---|---|---|---|---|---|---|---|---|---|
| | | **Our** | BayPy | LTM | BPC | **Our** | BayPy | LTM | BPC | **Our** | BayPy | LTM | BPC |
| $SM_1 + MM_1$ | 5k | 0.15(3) | 0.10(2) | 0.15(3) | 0.96(10) | 0.00(0) | 0.10(2) | 0.00(0) | 0.00(0) | 0.05(1) | 0.00(0) | 0.00(0) | 0.00(0) |
| | 10k | 0.05(1) | 0.05(1) | 0.10(2) | 0.90(10) | 0.00(0) | 0.05(1) | 0.00(0) | 0.00(0) | 0.00(0) | 0.00(0) | 0.00(0) | 0.00(0) |
| | 50k | 0.00(0) | 0.00(0) | 0.00(0) | 0.90(10) | 0.00(0) | 0.00(0) | 0.00(0) | 0.00(0) | 0.00(0) | 0.00(0) | 0.00(0) | 0.00(0) |
| $SM_2 + MM_1$ | 5k | 0.23(5) | 0.19(6) | 0.26(6) | 0.90(10) | 0.00(0) | 0.19(6) | 0.03(1) | 0.00(0) | 0.05(2) | 0.19(6) | 0.23(6) | 0.00(0) |
| | 10k | 0.13(4) | 0.13(4) | 0.13(4) | 0.86(10) | 0.00(0) | 0.03(4) | 0.00(0) | 0.00(0) | 0.00(0) | 0.13(4) | 0.13(4) | 0.00(0) |
| | 50k | 0.06(2) | 0.10(3) | 0.10(3) | 0.86(10) | 0.00(0) | 0.13(4) | 0.00(0) | 0.00(0) | 0.00(0) | 0.13(4) | 0.10(3) | 0.00(0) |
| $SM_2 + MM_2$ | 5k | 0.12(2) | 0.19(6) | 0.21(5) | 0.90(10) | 0.00(0) | 0.19(6) | 0.00(0) | 0.00(0) | 0.03(1) | 0.16(6) | 0.21(5) | 0.00(0) |
| | 10k | 0.03(1) | 0.13(4) | 0.10(3) | 0.86(10) | 0.00(0) | 0.13(4) | 0.00(0) | 0.00(0) | 0.00(0) | 0.11(4) | 0.10(3) | 0.00(0) |
| | 50k | 0.00(0) | 0.07(2) | 0.07(2) | 0.83(10) | 0.00(0) | 0.07(2) | 0.00(0) | 0.00(0) | 0.00(0) | 0.07(2) | 0.06(2) | 0.00(0) |
| $SM_3 + MM_1$ | 5k | 0.25(6) | 0.30(6) | 0.55(10) | 0.86(10) | 0.00(0) | 0.30(6) | 0.00(0) | 0.00(0) | 0.12(5) | 0.20(6) | 0.55(10) | 0.00(0) |
| | 10k | 0.17(5) | 0.25(5) | 0.50(10) | 0.83(10) | 0.00(0) | 0.25(5) | 0.00(0) | 0.00(0) | 0.05(3) | 0.16(5) | 0.50(10) | 0.00(0) |
| | 50k | 0.08(3) | 0.20(4) | 0.50(10) | 0.83(10) | 0.00(0) | 0.20(4) | 0.00(0) | 0.00(0) | 0.03(2) | 0.13(4) | 0.50(10) | 0.00(0) |

use the characteristic root statistic, abbreviated as CR statistic Robin and Smith (2000), to test the rank of the probability contingency matrix of two observed variables. Specifically, Let $\tilde{M}$ be an asymptotically normal estimator of $M$, then the CR statistic is the sum of $d - r$ smallest singular values of $\tilde{M}$, multiplied by the sample size. Under the null hypothesis, the above statistic converges in distribution to a weighted (given by the eigenvalues) sum of independent $\mathcal{X}_1^2$ random variables.

**Goodness-of-fit test for tensor rank.** Once the dimension of the support of latent variables is identified, in the structure learning procedure, we perform the following hypotheses test: $\mathcal{H}_0$: $\text{Rank}(\mathcal{T}) = r$ *v.s.* $\mathcal{H}_1$: $\text{Rank}(\mathcal{T}) \neq r$. To achieve this, we first apply the canonical polyadic (CP) decomposition technology to the target tensor $\mathcal{T}$ as a sum of $r$ rank-one tensors given specified $r$, then we evaluate how well the reconstructed tensor from this decomposition approximates the original tensor to conduct the hypotheses test.

To perform the rank-decomposition with specified $r$ on the probability contingency tensor, one can use the non-negative CP decomposition to decompose the tensor into the sum of $r$ rank-one tensor Shashua and Hazan (2005). Given the decomposition, one can obtain a reconstructed tensor, denoted by $\tilde{\mathcal{T}}$, from the outer product of decomposed vectors.

With the reconstructed tensor, we constructed square-chi goodness of fit test Cochran (1952) for testing $\text{Rank}(\mathcal{T}) = r$. Such a test is frequently used to summarize the discrepancy between observed values and the expected values, which measure the sum of differences between observed and expected outcome frequencies. Let $\mathbf{vec}(\mathcal{T})$ be the vectorization of tensor $\mathcal{T}$, suppose $\mathbf{vec}(\tilde{\mathcal{T}})$ be the asymptotic normality estimator of $\mathbf{vec}(\mathcal{T})$, we have the chi-square statistic as $\mathcal{X}^2 = \sum_{i \in \mathbf{vec}(\tilde{\mathcal{T}})} \frac{(\mathbf{vec}(\mathcal{T})_i - \mathbf{vec}(\tilde{\mathcal{T}})_i)^2}{\mathbf{vec}(\tilde{\mathcal{T}})_i}$, which follows the $\mathcal{X}^2$ distribution with freedom degrees $\prod_{i \in [n]} d_i - (\sum_{i,j \in [n]} d_i d_j)$.

## 5 Simulation Studies

In this section, we conducted simulation studies to assess the correctness of the proposed methods. The baseline approaches include Building Pure Cluster (BPC) Silva et al. (2006), Latent Tree Model (LTM) Choi et al. (2011), and Bayesian Pyramid Model (BayPy) Gu and Dunson (2023).

In the following simulation studies, we consider the different combinations of various types of structure models(SM) and measurement models(MM). Specifically, for the structure model, we consider the following five typical cases: [SM1]: $L_1 \rightarrow L_2$; [SM2]: $L_1 \rightarrow L_2 \rightarrow L_3$; [SM3]: the structure of latent variables is shown in Fig. 2(a); [Collider]: $L_1 \rightarrow L_2 \leftarrow L_3$; [Star]: $L_1 \rightarrow L_2, L_1 \rightarrow L_3, L_1 \rightarrow L_4$. For the measurement model, we consider the following two cases: [MM1]: each latent variable has three pure observed variables, i.e., $L_i \rightarrow \{X_1, X_2, X_3\}$; [MM2]: each latent variable has four pure observed variables, i.e., $L_i \rightarrow \{X_1, X_2, X_3, X_4\}$.

In all cases, the data generation process follows the discrete 3PLSM model: (i) we generate the probability contingency table of latent variables in advance, according to different latent structures (e.g., SM1), then (ii) we generate the conditional contingency table of observed variables (condition on their latent parent), and finally (iii) we sample the observed data according to the probability contingency table, where the dimension of latent support $r$ is set to 3 and the dimension of all observed variables support is set to 4, sample size ranged from $\{5k, 10k, 50k\}$.

Table 3: Results on learning the structure model. The symbol '-' indicates that the current method does not output this information. Lower value means higher accuracy.

| Algorithm | | Edge omission | | | | Edge commission | | | | Orientation omission | | | |
| --- | --- | --- | --- | --- | --- | --- | --- | --- | --- | --- | --- | --- | --- |
| | | **Our** | BayPy | LTM | BPC | **Our** | BayPy | LTM | BPC | **Our** | BayPy | LTM | BPC |
| $Collider+MM_1$ | 5k | 0.00(0) | 1.00(10) | 0.26(8) | 1.00(10) | 0.10(1) | 0.00(0) | 0.00(0) | 0.00(0) | 0.10(1) | 1.00(10) | – | 1.00(0) |
| | 10k | 0.00(0) | 1.00(10) | 0.23(6) | 1.00(10) | 0.00(0) | 0.02(1) | 0.0(0) | 0.00(0) | 0.00(0) | 1.00(10) | – | 1.00(0) |
| | 50k | 0.00(0) | 1.00(10) | 0.10(3) | 1.00(10) | 0.00(0) | 0.00(0) | 0.00(0) | 0.00(0) | 0.00(0) | 1.00(10) | – | 1.00(0) |
| $SM_2 + MM_1$ | 5k | 0.15(3) | 1.00(10) | 0.16(6) | 1.00(10) | 0.10(1) | 0.00(0) | 0.00(0) | 0.00(0) | 0.00(0) | 0.00(0) | – | 0.00(0) |
| | 10k | 0.05(1) | 1.00(10) | 0.13(4) | 1.00(10) | 0.01(1) | 0.00(0) | 0.00(0) | 0.00(0) | 0.00(0) | 0.00(0) | – | 0.00(0) |
| | 50k | 0.00(0) | 1.00(10) | 0.10(3) | 1.00(10) | 0.00(0) | 0.00(0) | 0.00(0) | 0.00(0) | 0.00(0) | 0.00(0) | – | 0.00(0) |
| $Star + MM_1$ | 5k | 0.10(3) | 1.00(10) | 0.25(5) | 1.00(10) | 0.20(5) | 0.00(0) | 0.00(0) | 0.00(0) | 0.00(0) | 0.00(0) | – | 0.00(0) |
| | 10k | 0.06(2) | 1.00(10) | 0.15(3) | 1.00(10) | 0.08(3) | 0.00(0) | 0.00(0) | 0.00(0) | 0.00(0) | 0.00(0) | – | 0.00(0) |
| | 50k | 0.03(1) | 1.00(10) | 0.15(3) | 1.00(10) | 0.05(2) | 0.00(0) | 0.00(0) | 0.00(0) | 0.00(0) | 0.00(0) | – | 0.00(0) |
| $SM_3 + MM_1$ | 5k | 0.22(7) | 1.00(10) | 0.50(10) | 1.00(10) | 0.40(6) | 0.00(0) | 0.02(1) | 0.00(0) | 0.20(2) | 1.00(10) | – | 1.00(10) |
| | 10k | 0.15(5) | 1.00(10) | 0.50(10) | 1.00(10) | 0.10(2) | 0.00(0) | 0.00(0) | 0.00(0) | 0.10(1) | 1.00(10) | – | 1.00(10) |
| | 50k | 0.05(2) | 1.00(10) | 0.50(10) | 1.00(10) | 0.05(1) | 0.00(0) | 0.00(0) | 0.00(0) | 0.00(0) | 1.00(10) | – | 1.00(10) |

For each simulation study, we randomly generate the dataset and apply the proposed algorithm and baselines to these data. We use the following scores for evaluating the performance of causal clusters from each algorithm: **latent omission**, **latent commission**, and **mismeasurement**. Moreover, to assess the ability of these algorithms to correctly discover the causal structure among latent variables, we use the metric like **edge omission (EO)**, **edge commission (EC)**, and **orientation omission (OO)**. These metrics can be referred to Silva et al. (2006), in which the tasks are aligned with our work. Each experiment was repeated ten times with randomly generated data, and the results were averaged.

The results are reported in Table 2 and Table 3. Our method consistently delivers the best outcomes across most scenarios, demonstrating its capability to identify both the causal clusters and the causal structures of latent variables. In contrast, the BPC approach performs poorly, as it is specifically designed for linear models. Additionally, the LTM and BayPy algorithms show suboptimal performance in structure learning of latent variables due to their limitations to specific structural models, such as tree structures, or assumptions that latent variables are binary.

# 6 Discussion and Further Work

The preceding sections presented the identification of discrete 3PLSM. In this section, we examine whether the impure structure (e.g., an edge between observed variables) can be detected through tensor rank conditions. For instance, consider the structure shown in Fig. 3 and suppose all observed variable has the same support. One can observe that for any subset $\{X_i, X_j, X_k\} \subset$ $\{X_1, X_2, X_3, X_4, X_5\}$ where $\{X_4, X_5\} \not\subset \{X_i, X_j, X_k\}$, one has

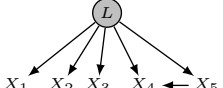

Figure 3: Example of identifying the impure structure.

$\text{Rank}(\mathcal{T}_{(X_i, X_j, X_k)}) = |\text{supp}(L)|$. This implies $\{X_1, X_2, X_3\}$ and $X_4$ (or $X_5$) share the common latent parent. Moreover, we have $\text{Rank}(\mathcal{T}_{(X_4, X_5)}) = |\text{supp}(X_5)|$, assuming the cardinality of the observed variables is larger than that of the latent variables. By the graphical criteria of the tensor rank condition, one can infer that there is an edge between $X_4$ and $X_5$, indicating the impure structure can be identified by the tensor rank condition. Developing an efficient algorithm to learn a more general discrete 3PLSM structure that allows impure structure in a principled way is part of our future work.

# 7 Conclusion

We derive a nontrivial algebraic property of a particular type of discrete causal model under proper causal assumptions. We build the connection between tensor rank and the d-separation relations in the causal graph and propose the graphical criteria of tensor rank. By this, the identifiability of causal structure in discrete latent structure models is achieved based on which we proposed an identification to locate latent causal variables and identify their causal structure. We provide a practical test approach for testing the tensor rank and verifying the efficientness of the proposed algorithm via the simulated studies. The proposed theorems and the algorithms take a meaningful step in understanding the causal mechanism of discrete data. However, the proposed method can hardly be applied to high-dimensional discrete data and only applies to pure-children structures. Therefore, relaxing these restrictions and making them scalable to high-dimensional real-world datasets and more general structural constraints would be a meaningful future direction.

## Acknowledgments

This research was supported in part by National Key R&D Program of China (2021ZD0111501), National Science Fund for Excellent Young Scholars (62122022), Natural Science Foundation of China (61876043, 61976052, 62476163), the major key project of PCL (PCL2021A12). ZM would like to acknowledge the support of the China Scholarship Council (CSC). FX would like to acknowledge the support by the Natural Science Foundation of China (62306019). JQ would like to acknowledge the support by the Natural Science Foundation of China (62406080). KZ would like to acknowledge the support from NSF Award No. 2229881, AI Institute for Societal Decision Making (AI-SDM), the National Institutes of Health (NIH) under Contract R01HL159805, and grants from Salesforce, Apple Inc., Quris AI, and Florin Court Capital. We appreciate the comments from anonymous reviewers and Area Chairs, which greatly helped to improve the paper.

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

## Supplementary Material

The supplementary material contains

- Graphical Notations;
- Example of Tensor Representations of Joint Distribution;
- Discussion of Our Assumptions;
- Proofs of Main Results;
  - Proof of Theorem 3.3;
  - Proof of Proposition 4.2;
  - Proof of Proposition 4.3;
  - Proof of Proposition 4.5;
  - Proof of Theorem 4.6;
  - Proof of Theorem 4.7;
  - Proof of Theorem 4.9.
- Extension of Different Latent State Space;
- Discussion with the Hierarchical Structures;
- Practical Estimation of Tensor Rank;
- More Experimental Results;
- Experimental Results on Real-world Dataset;

## A   Graphical Notations

Below, we provide some graphical notation used in our work, which is mainly derived from the Pearl (2009); Spirtes et al. (2000).

**Definition A.1** (Path and Directed Path)**.** *In a DAG, a **path** $P$ is a sequence of nodes $(V_1, ...V_r)$ such that $V_i$ and $V_{i+1}$ are adjacent in $\mathcal{G}$, where $1 \le i < r$. Further, we say a path $P = (V_{i_0}, V_{i_1}, \ldots, V_{i_k})$ in $G$ is a **directed path** if it is a sequence of nodes of $G$ where there is a directed edge from $V_{i_j}$ to $V_{i_{(j+1)}}$ for any $0 \le j \le k - 1$.*

**Definition A.2** (Collider)**.** *A **collider** on a path $\{V_1, ...V_p\}$ is a node $V_i$ , $1 < i < p$, such that $V_{i-1}$ and $V_{i+1}$ are parents of $V_i$.*

Graphically, we also say a collider is a 'V-structure'.

**Definition A.3** (d-separation)**.** *A path $p$ is said to be d-separated (or blocked) by a set of nodes $\mathbf{Z}$ if and only if the following two conditions hold:*

- *$p$ contains a chain $V_i \rightarrow V_k \rightarrow V_j$ or a fork $V_i \leftarrow V_k \rightarrow V_j$ such that the middle node $V_k$ is in $\mathbf{Z}$;*

- *$p$ contains a collider $V_i \rightarrow V_k \leftarrow V_j$ such that the middle node $V_k$ is not in $\mathbf{Z}$ and such that no descendant of $V_k$ is in $\mathbf{Z}$.*

A set $\mathbf{Z}$ is said to d-separate $\mathbf{A}$ and $\mathbf{B}$ if and only if $\mathbf{Z}$ blocks every path from a node in $\mathbf{A}$ to a node in $\mathbf{B}$. We also denote as $\mathbf{A} \perp\!\!\!\perp \mathbf{B}|\mathbf{Z}$ in the causal graph model.

## B   Example of Tensor Representations of Joint Distribution

Consider a single latent variable structure that has three pure observed variables, i.e., $L_1 \rightarrow \{X_1, X_2, X_3\}$. We aim to illustrate the tensor representation of the probability contingency table and the tensor rank condition for the joint distribution $\mathbb{P}(X_1 X_2 X_3)$. For convenience, let $\text{supp}(L_1) = \{0, 1\}$ and $\text{supp}(X_i) = \{0, 1, 2\}$. We further denote $p_{ij} = \mathbb{P}(X_1 = i, X_2 = j)$, $p_{\hat{i}|j} = \mathbb{P}(X_1 = i|L_1 = j)$, $p_{\tilde{i}|j} = \mathbb{P}(X_2 = i|L_1 = j)$, $\bar{p}_i = \mathbb{P}(L_1 = i)$, and $p_{ijk} = \mathbb{P}(X_1 = i, X_2 = j, X_3 = k)$. For the joint distribution of $\mathbb{P}(X_1 X_2)$, we have the tensor representation as follows:

$$\mathcal{T}_{(X_1 X_2)} = \begin{bmatrix} p_{00} & p_{01} & p_{02} \\ p_{10} & p_{11} & p_{12} \\ p_{20} & p_{21} & p_{22} \end{bmatrix} = \underbrace{\begin{bmatrix} p_{\hat{0}|0} & p_{\hat{0}|1} \\ p_{\hat{1}|0} & p_{\hat{1}|1} \\ p_{\hat{2}|0} & p_{\hat{2}|1} \end{bmatrix}}_{\mathcal{T}_{(X_1|L_1)}} \cdot \underbrace{\begin{bmatrix} \bar{p}_0 & 0 \\ 0 & \bar{p}_1 \end{bmatrix}}_{\mathrm{Diag}(\mathcal{T}_{(L_1)})} \cdot \underbrace{\begin{bmatrix} p_{\tilde{0}|0} & p_{\tilde{1}|0} & p_{\tilde{2}|0} \\ p_{\tilde{0}|1} & p_{\tilde{1}|2} & p_{\tilde{2}|1} \end{bmatrix}}_{\mathcal{T}_{(X_2|L_1)}^{\mathsf{T}}}, \tag{1}$$

where $\mathcal{T}_{(X_1|L_1)}$ is the tensor representation of $\mathbb{P}(X_1|L_1)$, $\mathcal{T}_{(X_2|L_1)}$ is the tensor representation of $\mathbb{P}(X_2|L_1)$ and $\mathrm{Diag}(\mathcal{T}_{(L_1)})$ is the diagonalization of $\mathbb{P}(L_1)$.

Under the Full Rank assumption, we have $\mathcal{T}_{(X_1|L_1)}$ and $\mathcal{T}_{(X_2|L_1)}$ are column full rank. Thus, the rank of $\mathcal{T}_{X_1 X_2}$ is two, i.e., $\mathrm{Rank}(\mathcal{T}_{X_1 X_2}) = |\mathrm{supp}(L_1)| = 2$. This illustrate the Prop. 1.

Next, we consider the three-way tensor $\mathcal{T}_{(X_1 X_2 X_3)}$ of the joint distribution $\mathbb{P}(X_1 X_2 X_3)$. We will represent the three-way tensor as its frontal slices Kolda and Bader (2009), i.e., three matrices for $\mathbb{P}(X_1 X_2 X_3 = 0)$, $\mathbb{P}(X_1 X_2 X_3 = 1)$ and $\mathbb{P}(X_1 X_2 X_3 = 2)$.

$$\underbrace{\begin{bmatrix} p_{000} & p_{010} & p_{020} \\ p_{100} & p_{110} & p_{120} \\ p_{200} & p_{210} & p_{220} \end{bmatrix}}_{\mathcal{T}_{(X_1 X_2 X_3 = 0)}}, \underbrace{\begin{bmatrix} p_{001} & p_{011} & p_{021} \\ p_{101} & p_{111} & p_{121} \\ p_{201} & p_{211} & p_{221} \end{bmatrix}}_{\mathcal{T}_{(X_1 X_2 X_3 = 1)}}, \underbrace{\begin{bmatrix} p_{002} & p_{012} & p_{022} \\ p_{102} & p_{112} & p_{122} \\ p_{202} & p_{212} & p_{222} \end{bmatrix}}_{\mathcal{T}_{(X_1 X_2 X_3 = 2)}}. \tag{2}$$

In the contingency tensor above, the element of $\mathcal{T}_{(X_1 X_2 X_3)}$ is

$$\begin{aligned}
&\mathbb{P}(X_1 = i, X_2 = j, X_3 = k) \\
&= \sum_{r=0}^{1} \mathbb{P}(X_1 = i, X_2 = j, X_3 = k | L_1 = r)\mathbb{P}(L_1 = r) \\
&= \sum_{r=0}^{1} \mathbb{P}(X_1 = i | L_1 = r)\mathbb{P}(X_2 = j | L_1 = r)\mathbb{P}(X_3 = k | L_1 = r)\mathbb{P}(L_1 = r).
\end{aligned} \tag{3}$$

One can represent the tensor $\mathcal{T}_{X_1 X_2 X_3 | L_1 = r}$ as

$$\mathcal{T}_{(X_1 X_2 X_3 | L_1 = r)} = \begin{bmatrix} \mathbb{P}(X_1 = 0 | L_1 = r) \\ \mathbb{P}(X_1 = 1 | L_1 = r) \\ \mathbb{P}(X_1 = 2 | L_1 = r) \end{bmatrix} \otimes \begin{bmatrix} \mathbb{P}(X_2 = 0 | L_1 = r) \\ \mathbb{P}(X_2 = 1 | L_1 = r) \\ \mathbb{P}(X_2 = 2 | L_1 = r) \end{bmatrix} \otimes \begin{bmatrix} \mathbb{P}(X_3 = 0 | L_1 = r) \\ \mathbb{P}(X_3 = 1 | L_1 = r) \\ \mathbb{P}(X_3 = 2 | L_1 = r) \end{bmatrix}, \tag{4}$$

where $\otimes$ represent the outer product, e.g., for two vector $\mathbf{u}$ and $\mathbf{v}$, $(\mathbf{u} \otimes \mathbf{v})_{ij} = u_i v_j$ with $\mathbf{u} = \{u_1, \cdots, u_n\}$ and $\mathbf{v} = \{v_1, \cdots, v_n\}$.

According to the definition of tensor rank, one can see that $\mathcal{T}_{(X_1 X_2 X_3 | L_1 = r)}$ is a rank-one tensor. Furthermore, since $\mathcal{T}_{(X_1 X_2 X_3)} = \sum_{r=1}^{2} \mathcal{T}_{(X_1 X_2 X_3 | L_1 = r)} \mathcal{T}_{(L_1 = r)}$, the rank of $\mathcal{T}_{(X_1 X_2 X_3)}$ is two under Assumption 2.2 ~ Assumption 2.4. This is because $L_1$ d-separates $X_i$ and $X_j$ with $\forall i, j \in [1, 2, 3]$, which illistrate the graphical criteria of tensor rank condition.

## C   Discussion of Our Assumptions

To study the discrete statistical model, certain commonly-used parameter assumptions are necessary. For instance, the Full Rank assumption (Assumption 2.4), as utilized in our study, ensures diversity within the parameter space. This is crucial to prevent the contingency table of the joint distribution from collapsing into a lower-dimensional space. There are related works that also use such an assumption Leonard and Novick (1986); Bartolucci et al. (2007); Gu (2022). Essentially, in our work, such an assumption ensures the effectiveness of rank decomposition (e.g., minimal decomposition or unique decomposition Kruskal (1977)), which induces the structural identifiability of a discrete causal model. Moreover, the sufficient observation assumption that the dimension of latent support is larger than the dimension of observed support is reasonable, as it ensures sufficient measurement of the latent variable. We also discuss this assumption in the remark E.4 that demonstrates the CI relation is testable if this assumption holds.

It is worth noting that, although Kivva et al. (2021) shows that the causal structure among latent variables can be identifiable under weaker assumptions, by leveraging the identifiability of the mixture model, we emphasize that our algorithm, based on the tensor rank condition, is simpler and more efficient. It can also produce more robust results when performing causal discovery on the observational data. Moreover, in Kivva et al. (2021), the parameters of the mixture oracle, such as the number of components, are estimated using approximate methods like the K-means algorithm, which may not be applied directly to discrete data.

## D    Proofs of Main Results

### D.1    Proof of Theorem 3.3

*Proof.* "If" part:

We prove this by contradiction, i.e., suppose (i). there exists a variable set $\mathbf{S}$ in the causal graph $\mathcal{G}$ with $|\text{supp}(\mathbf{S})| = r$ that d-separates any pair of variables in $\{X_1, \cdots, X_n\}$, and (ii).does no exist conditional set $\tilde{\mathbf{S}}$ that satisfies $|\text{supp}(\tilde{\mathbf{S}})| < r$, then $\text{Rank}(\mathcal{T}_{\{X_1 \dots X_n\}}) \neq r$. There are two cases we need to consider: Case 1: $\text{Rank}(\mathcal{T}_{\{X_1 \dots X_n\}}) > r$, and Case 2: $\text{Rank}(\mathcal{T}_{\{X_1 \dots X_n\}}) < r$.

Case 1: Due to $\mathbf{S}$ is conditional set that d-separates all variables in $\{X_1, \cdots, X_n\}$, then we have $\mathbb{P}(X_1 \cdots X_n) = \sum_{i=1}^{r} (\prod_{j=1}^{n} \mathbb{P}(X_j|\mathbf{S} = i))\mathbb{P}(\mathbf{S} = i)$. When $\text{Rank}(\mathcal{T}_{\{X_1 \dots X_n\}}) > r$, let $\text{Rank}(\mathcal{T}_{\{X_1 \dots X_n\}}) = k > r$ ,there are

$$
\begin{aligned}
&\mathcal{T}_{\{X_1 \dots X_n\}} \\
&= \sum_{i=1}^{r} \mathcal{T}_{(X_1|\mathbf{S}=i)} \otimes \cdots \otimes \mathcal{T}_{(X_n|\mathbf{S}=i)}\mathcal{T}_{(\mathbf{S}=i)} \\
&= \sum_{j=1}^{k} \mathbf{u}_1^{(j)} \otimes \cdots \otimes \mathbf{u}_n^{(j)},
\end{aligned}
\tag{5}
$$

which violates the definition of tensor rank (i.e., $k$ is not a minimal rank-one decomposition, it can be reduced to the smaller decomposition with $r$). Meanwhile, if there exists other conditional sets $\tilde{\mathbf{S}}$ such that $\mathbb{P}(X_1 \cdots X_n) = \sum_{i=1}^{k} (\prod_{j=1}^{n} \mathcal{T}_{(X_j|\tilde{\mathbf{S}}=i)})\mathcal{T}_{(\tilde{\mathbf{S}}=i)}$ is minimal rank decomposition, it violates the condition (ii) that $|\text{supp}(\mathbf{S})|$ is minimal. ,

Case 2: When $\text{Rank}(\mathcal{T}_{\{X_1 \dots X_n\}}) < r$, let $\text{Rank}(\mathcal{T}_{\{X_1 \dots X_n\}}) = t < r$, due to $\mathbf{S}$ is conditional set with smallest support $r$ in the causal graph, one have

$$
\begin{aligned}
&\mathcal{T}_{\{X_1 \dots X_n\}} \\
&= \sum_{i=1}^{r} \mathcal{T}_{(X_1|\mathbf{S}=i)} \otimes \cdots \otimes \mathcal{T}_{(X_n|\mathbf{S}=i)}\mathcal{T}_{(\mathbf{S}=i)} \\
&= \sum_{j=1}^{t} \mathbf{u}_1^{(j)} \otimes \cdots \otimes \mathbf{u}_n^{(j)},
\end{aligned}
\tag{6}
$$

which means that there at least exist $\mathcal{T}_{(X_1 \dots X_n|\mathbf{S}=i)}$ and $\mathcal{T}_{(X_1 \dots X_n|\mathbf{S}=j)}$ such that $\mathcal{T}_{(X_j|\mathbf{S}=i)} = \alpha\mathcal{T}_{(X_k|\mathbf{S}=i)}$ for any $i, k \in [n]$ where $\alpha$ is a constant, i.e, the columns of the conditional contingency table are linearly dependent. This violates the assumption 2.4 (the Full Rank assumption).

Therefore, $\text{Rank}(\mathcal{T}_{\{X_1 \dots X_n\}}) = r$ if the condition (i) and condition (ii) holds.

"Only if" part:

We will show if one of the conditions is violated, the tensor rank is not $r$, i.e., if (i). there does not exist a variable set $\mathbf{S}$ in the causal graph with $|\text{supp}(\mathbf{L})| = r$ that d-separates any pair of variables in $\{X_1, \cdots, X_n\}$, or (ii). exist $\tilde{\mathbf{S}}$ that satisfies $|\text{supp}(\tilde{\mathbf{S}})| < r$, then $\text{Rank}(\mathcal{T}_{\{X_1 \dots X_n\}}) \neq r$.

We first show the case that condition (i) is violated. There are two cases we need to consider, i.e., Case 1: $\mathbf{S}$ is not a conditional set in the causal graph, and Case 2: $\mathbf{S}$ is a variable constructed from parameter space.

Case 1: if $\mathbf{S}$ is not a conditional set and $\mathbf{S} \cap \mathrm{Des}_{X_1} \cap \cdots \cap \mathrm{Des}_{X_n} = \varnothing$, by the Markov assumption, $\mathbb{P}(X_1, \cdots, X_n | \mathbf{S} = i) \neq \prod_{j=1}^{n} \mathbb{P}(X_j | \mathbf{S} = i)$. By Lemma. D.3, $\mathbb{P}(X_1, \cdots, X_n | \mathbf{S} = i)$ is not a rank-one tensor, which violates the definition of tensor rank.

If $\mathbf{S}$ is not a conditional set and $\mathbf{S} \cap \mathrm{Des}_{X_1} \cap \cdots \cap \mathrm{Des}_{X_n} \neq \varnothing$, we will show that $\mathbb{P}(X_1, \cdots, X_n | \mathbf{S} = i)$ is not a rank-one tensor, to prove such a rank-decomposition does not exist. Under the faithfulness assumption and Markov assumption, let $\tilde{\mathbf{S}}$ be the a minimal conditional set with $|\mathrm{supp}(\tilde{\mathbf{S}})| = k$ for any pair variable in $\{X_1, \cdots, X_n\}$, we have

$$
\begin{aligned}
&\mathcal{T}_{(X_1 \cdots X_n | \mathbf{S}=i)} \\
&= \sum_{j=1}^{k} \mathcal{T}_{(X_1 \cdots X_n | \mathbf{S}=i, \tilde{\mathbf{S}}=j)} \mathcal{T}_{(\mathbf{S}=i | \tilde{\mathbf{S}}=j)}.
\end{aligned}
\tag{7}
$$

If $\mathcal{T}_{(X_1 \cdots X_n | \mathbf{S}=i)}$ is a rank-one tensor, i.e., $\mathcal{T}_{(X_1 \cdots X_n | \mathbf{S}=i)} = \mathbf{u}_1 \otimes \cdots \otimes \mathbf{u}_n$, we further have

$$
\sum_{j=1}^{k} \mathcal{T}_{(X_1 \cdots X_n | \mathbf{S}=i, \tilde{\mathbf{S}}=j)} \mathcal{T}_{(\mathbf{S}=i | \tilde{\mathbf{S}}=j)} = \mathbf{u}_1 \otimes \cdots \otimes \mathbf{u}_n.
\tag{8}
$$

Note that if there exists $X_p, X_q \in \{X_1, \cdots, X_n\}$ such that $\mathbf{S}$ is the common descendant variable of $X_p$ and $X_q$, it will lead to a collider structure in which $\mathbb{P}(X_i | \mathbf{S})$ and $\mathbb{P}(X_j | \mathbf{S})$ is relevant (i.e., the v-structure is activated). Thus, let $\mathbf{X}_t = \{X_1, \cdots, X_n\} \setminus \{X_i, X_j\}$, $\mathcal{T}_{(X_1 \cdots X_n | \mathbf{S}=i)}$ is not a rank-one tensor due to the sub-tensor $\mathcal{T}_{(X_i X_j, \mathbf{X}_t = \mathbf{c} | \mathbf{S}=i)}$ (a slice of tensor $\mathcal{T}_{(X_1 \cdots X_n | \mathbf{S}=i)}$) is not a rank-one tensor [2]Hackbusch (2012); Kruskal (1977), under the faithfulness assumption. If so, one have $\mathcal{T}_{(X_i X_j, \mathbf{X}_t = \mathbf{c} | \mathbf{S}=i)} = \mathbf{u}_1 \otimes \mathbf{u}_2$. Let $\mathbf{u}_1 = \mathbb{P}(X_i, \mathbf{X}_t = \mathbf{c} | \mathbf{S} = i)$ and $\mathbf{u}_2 = \mathbb{P}(X_j, \mathbf{X}_t = \mathbf{c} | \mathbf{S} = i)$, it violates the faithfulness assumption.

Thus, $\mathbf{S}$ can not be the common descendant of any pair variables in $\{X_1, \cdots, X_n\}$. So, for any $X_p, X_q \in \{X_1, \cdots, X_n\}$, there are $\mathcal{T}_{(X_p X_q | \mathbf{S}=j \tilde{\mathbf{S}}=i)} = \mathcal{T}_{(X_p | \tilde{\mathbf{S}}=i, \mathrm{Des}_{X_p})=c} \otimes \mathcal{T}_{(X_q | \tilde{\mathbf{S}}=i, \mathrm{Des}_{X_q})=c}$ according to Markov assumption ($\tilde{\mathbf{S}}$ is a d-separation set for $X_p$ and $X_q$).

Now, for the Eq. 8, we have

$$
\begin{aligned}
&\mathcal{T}_{(X_1 \cdots X_n | \mathbf{L}=i)} \\
&= \sum_{j=1}^{k} \otimes_{t=1}^{n} \mathcal{T}_{(X_t | \tilde{\mathbf{S}}=j, \mathrm{Des}_{X_t}=c)} \mathcal{T}_{(\tilde{\mathbf{S}}=j, \mathrm{Des}_{X_t}=c | \tilde{\mathbf{S}}=j)} \\
&= \mathbf{u}_1 \otimes \cdots \otimes \mathbf{u}_n.
\end{aligned}
\tag{9}
$$

This equality holds if the $k$ sum of the rank-one tensor can be reduced to a rank-one tensor, i.e., for any $X_p$, $\mathcal{T}_{(X_p | \tilde{\mathbf{S}}=1, \mathrm{Des}_{X_p}=c)} = \alpha_2 \mathcal{T}_{(X_p | \tilde{\mathbf{S}}=2, \mathrm{Des}_{X_p}=c)} = \cdots = \alpha_r \mathcal{T}_{(X_p | \tilde{\mathbf{S}}=r, \mathrm{Des}_{X_p}=c)}$, where $\alpha_i$ and $c$ are constant. However, this equality can not hold due to the following reasons.

If $\mathrm{Pa}_{X_p} \neq \varnothing$, let $L_p$ be the parent of $X_p$, we have

$$
\begin{aligned}
&\mathcal{T}_{(X_p | \tilde{\mathbf{S}}=i, \mathrm{Des}_{X_p}=c)} \\
&= \sum_{j=1}^{|\mathrm{supp}(L_p)|} \mathcal{T}_{(X_p | L_p=j, \mathrm{Des}_{X_p}=c)} \mathcal{T}_{(L_p=j | \tilde{\mathbf{S}}=i, \mathrm{Des}_{X_p}=c)} \\
&\neq \sum_{j=1}^{|\mathrm{supp}(L_p)|} \mathcal{T}_{(X_p | L_p=j, \mathrm{Des}_{X_p}=c)} \mathcal{T}_{(L_p=j | \tilde{\mathbf{S}}=r, \mathrm{Des}_{X_p}=c)} = \alpha \mathcal{T}_{(X_p | \tilde{\mathbf{S}}=r, \mathrm{Des}_{X_p}=c)},
\end{aligned}
\tag{10}
$$

---

[2]The lower bound of tensor rank is not less than the rank of any slice of tensor.

in which the inequality holds because of the Full Rank assumption and all marginal distribution probabilities are not zero (see Lemma. D.4). Therefore, $\mathcal{T}_{(X_p|\tilde{\mathbf{S}}=1,\mathrm{Des}_{X_p}=c)} \neq \cdots \neq \alpha_r \mathcal{T}_{(X_p|\tilde{\mathbf{S}}=r,\mathrm{Des}_{X_p}=c)}$. Thus, $\mathcal{T}_{(X_1 \cdots X_n|\mathbf{S}=i)}$ is not a rank-one tensor. By the definition of tensor rank, $\mathrm{Rank}(\mathcal{T}_{(X_1 \cdots X_n)}) \neq r$.

If $\mathrm{Pa}_{X_p} = \varnothing$, i.e., $X_p$ is the root variable, (e.g., $X_p \rightarrow \mathbf{S}$, $\mathrm{Des}_{X_p \in \mathbf{S}}$ ), we will show that the probability contingency table $\mathcal{T}_{(X_p|\mathbf{S})}$ is full rank, and then the equality in Eq. 8 can not hold. According to the Full Rank assumption and $X_p$ is the parent variable of $\mathbf{S}$, we have the probability contingency table $\mathcal{T}_{(\mathbf{S}|X_p)}$ is full rank.

Due to $\mathbb{P}(X_p, \mathbf{S}) = \mathbb{P}(\mathbf{S}|X_p)\mathbb{P}(X_p) = \mathbb{P}(X_p|\mathbf{S})\mathbb{P}(\mathbf{S})$ and the probability in the marginal distribution is not zero, we have $\mathcal{T}_{(X_p|\mathbf{S})} = \mathcal{T}_{(\mathbf{S}|X_p)}\mathrm{Diag}(\mathcal{T}_{(X_p)})\mathrm{Diag}(\mathcal{T}_{(\mathbf{S})})^\dagger$, where $\mathrm{Diag}(\mathcal{T}_{(X_p)})$ is a diagonalization of marginal distribution probability vector of $X_p$. One can see that $\mathcal{T}_{(X_p|\mathbf{S})}$ is full rank due to the diagonal matrices are all of full rank. Thus, $\mathcal{T}_{(X_p|\tilde{\mathbf{S}}=1)} \neq \cdots \neq \alpha_r \mathcal{T}_{(X_p|\tilde{\mathbf{S}}=r)}$ and $\mathcal{T}_{(X_1 \cdots X_n|\mathbf{S}=i)}$ is not a rank-one tensor. By the definition of tensor rank, $\mathrm{Rank}(\mathcal{T}_{(X_1 \cdots X_n)}) \neq r$.

Case 2: we further show that for the parameter space, the tensor $\mathcal{T}_{(X_1,\cdots,X_n)} = \sum_{i=1}^r \mathbf{u}_1 \otimes \cdots \otimes \mathbf{u}_n$ does not hold with $r$, where $\mathbf{u}_i$ represents any vector. Let $\mathbf{u}_i \otimes \cdots \otimes \mathbf{u}_n$ be a rank-one tensor of $\mathbb{P}(X_1,\cdots,X_n|\tilde{\mathbf{S}}=i)\mathbb{P}(\tilde{\mathbf{S}}=i)$ due to $\mathbb{P}(\tilde{\mathbf{S}}=i)$ is a constant. In other words, one can construct a variable set $\tilde{\mathbf{S}}$ with $|\mathrm{supp}(\tilde{\mathbf{S}})| = r$. For any $\tilde{\mathbf{S}}$ constructed from parameter space (i.e., $\tilde{\mathbf{S}}$ is not a true node set in the causal graph), if $\mathcal{T}_{(X_1,\cdots,X_n|\tilde{\mathbf{S}}=i)} = \mathbf{u}_1 \otimes \cdots \otimes \mathbf{u}_n$, one can let $\mathbf{u}_1 = \mathcal{T}_{(X_1|\tilde{\mathbf{S}}=i)}, \cdots, \mathbf{u}_n = \mathcal{T}_{(X_n|\tilde{\mathbf{S}}=i)}$, which violates the faithfulness assumption. Based on the above analysis, there does not exist $\tilde{\mathbf{S}}$ by any constructed such that $\mathcal{T}_{(X_1 \cdots X_n)}$ have the summation $r$ rank-one decomposition, i.e., $\mathrm{Rank}(\mathcal{T}_{(X_1 \cdots X_n)}) \neq r$.

Now, we analyze the condition (ii), i.e., there exists a conditional set $\tilde{\mathbf{S}}$ with $\mathrm{supp}(\tilde{\mathbf{S}}) < r$. Let $\mathrm{supp}(\tilde{\mathbf{S}}) = k$, $k < r$, we have $\mathcal{T}_{(X_1 \cdots X_n)} = \sum_{i=i}^k \mathcal{T}_{(X_1|\tilde{\mathbf{S}}=i)} \cdots \otimes \mathcal{T}_{(X_n|\tilde{\mathbf{S}}=i)}\mathcal{T}_{(\tilde{\mathbf{S}}=i)}$ is a smaller rank-one decomposition than $\mathcal{T}_{(X_1 \cdots X_n)} = \sum_{j=1}^r \mathbf{u^j}_1 \otimes \cdots \otimes \mathbf{u}_n^j$. According to the definition of tensor rank, we have $\mathrm{Rank}(\mathcal{T}_{\{X_1 \cdots X_n\}}) = k \neq r$.

In summary, the theorem is proven.

$\square$

**Lemma D.1.** *Let $L_p$ be the parent of $X_p$. Consider $\mathbb{P}(X_p|\mathbf{S}=i)$ and $\mathbb{P}(X_p|\mathbf{S}=r)$, where $\mathbf{S}$ can be any variable set. Under the Assumption 2.4, for a constant $c$, the following inequality holds:*

$$\sum_{j=1}^{|\mathrm{supp}(L_p)|} \mathcal{T}_{(X_p|L_p=j)}\mathcal{T}_{(L_p=j|\mathbf{S}=i)} \neq c \sum_{j=1}^{|\mathrm{supp}(L_p)|} \mathcal{T}_{(X_p|L_p=j)}\mathcal{T}_{(L_p=j|\mathbf{S}=r)}.$$

*Proof.* We prove it by contradiction. For convenience of symbols, let $\mathbf{v}_j = \mathcal{T}_{(X_p|L_p=j)}$, and $\alpha_{j|i} = \mathbb{P}(L_p = j|\mathbf{L} = i)$ and $\beta_{j|r} = \mathbb{P}(L_p = j|\mathbf{L} = r)$ $(\alpha_{j|i} \neq \beta_{j|r})$, if the equality hold, we have

$$\sum_{j=1}^{|\mathrm{supp}(L_p)|} \alpha_{j|i}\mathbf{v}_j = c \sum_{j=1}^{|\mathrm{supp}(L_p)|} \beta_{j|r}\mathbf{v}_j, \tag{11}$$

$\square$

which means that

$$\mathbf{v}_t = \frac{\sum_{k \neq t}^{|\mathrm{supp}(L_p)|}(\alpha_{k|i} - c\beta_{k|r})\mathbf{v}_k}{\alpha_{t|i} - c\beta_{t|r}}. \tag{12}$$

That is, $\mathbf{v}_t$ is a linear combination of other vectors $\mathbf{v}_k$ with $t \neq k$, i.e., the linear combination of other column vectors in the conditional probability contingency table, which is contrary to the Assumption 2.4.

**Lemma D.2.** *Let $L_p$ be the parent of $X_p$. Suppose Assumption 2.2 ~ Assumption 2.4 hold. $\mathcal{T}_{(X_p|\mathbf{S}=i)}$ cannot be expressed as a linear combination of other $q$ vectors $\mathcal{T}_{(X_p|\mathbf{S}=r)}$, i.e., $\mathcal{T}_{(X_p|\mathbf{S}=i)} \neq \sum_{r=1}^{q} \gamma_r \mathcal{T}_{(X_p|\mathbf{S}=r)}$.*

*Proof.* We prove it by contradiction. For convenience of symbols, let $\mathbf{v}_j = \mathcal{T}_{(X_p|L_p=j)}$, and $\alpha_{j|i} = \mathbb{P}(L_p = j|\mathbf{S} = i)$ and $\beta_{j|r} = \mathbb{P}(L_p = j|\mathbf{S} = r)$, if the equality hold, one have

$$
\begin{aligned}
\sum_{j=1}^{|\text{supp}(L_p)|} \alpha_{j|i}\mathbf{v}_j &= \sum_{r=1}^{q} \gamma_r \sum_{j=1}^{|\text{supp}(L_p)|} \beta_{j|r}\mathbf{v}_j \\
&= \sum_{j=1}^{|\text{supp}(L_p)|} \mathbf{v}_j \sum_{r=1}^{q} \gamma_r\beta_{j|r},
\end{aligned} \tag{13}
$$

there exist a vector $\mathbf{v}_t$ with $\alpha_{t|i} - \sum_{r=1}^{q} \gamma_r\beta_{t|r} \neq 0$, such that

$$
\mathbf{v}_t = \frac{\sum_{k\neq t}^{|\text{supp}(L_p)|} (\sum_{r=1}^{q} \gamma_r\beta_{j|r} - \alpha_{k|i})\mathbf{v}_k}{\alpha_{t|i} - \sum_{r=1}^{q} \gamma_r\beta_{t|r}}. \tag{14}
$$

It means $\mathcal{T}_{(X_p|L_p=t)}$ is a linear combination of other column vectors in the conditional probability contingency table, which is contrary to Assumption 2.4.

$\square$

**Lemma D.3.** *For the set of variables $\{X_1, \cdots, X_n\}$, if $\tilde{\mathbf{S}}$ is not a conditional set that d-separates any pair of variables in $\{X_1, \cdots, X_n\}$ in the causal graph and $\tilde{\mathbf{S}} \cap \text{Des}_{\{X_1\cdots X_n\}} = \varnothing$, then $\mathcal{T}_{(X_1\cdots X_n|\tilde{\mathbf{S}}=j)}$ is not a rank-one tensor.*

*Proof.* Let $\mathbf{S}$ be the minimal conditional set, (e.g., $\mathbf{S} = \{X_1, \cdots, X_{n-1}\}$), denote $|\text{supp}(\mathbf{S})| = r$, under the faithfulness assumption and the Markov assumption, $\mathbb{P}(\mathbf{S}) \neq \mathbb{P}(\tilde{\mathbf{S}})$, if $\mathcal{T}_{(X_1\cdots X_n|\tilde{\mathbf{S}}=j)}$ is a rank-one tensor, we have

$$
\begin{aligned}
&\mathcal{T}_{(X_1\cdots X_n|\tilde{\mathbf{S}}=j)} \\
&= \sum_{i=1}^{r} \otimes_{t=1}^{n} \mathcal{T}_{(X_t|\mathbf{S}=i)}\mathcal{T}_{(\mathbf{S}=i|\tilde{\mathbf{S}}=j)} \\
&= \mathbf{u}_1 \otimes \cdots \otimes \mathbf{u}_n,
\end{aligned} \tag{15}
$$

which means that for any $X_p \in \{X_1, \cdots X_n\}$, $\mathcal{T}_{(X_p|\mathbf{S}=1)} = \alpha_2\mathcal{T}_{(X_p|\mathbf{S}=2)} = \cdots \mathcal{T}_{(X_p|\mathbf{S}=r)}$.

If $X_p$ has a parent variable $L_p$ in the causal graph, by Lemma .D.1, the equality does not hold. Thus, $\mathcal{T}_{(X_1\cdots X_n|\tilde{\mathbf{L}}=j)}$ is not a rank-one tensor.

If $X_p$ is root node in the causal graph, and $\mathbf{S}$ is conditional set that d-separates $X_p$ and $\{X_1, \cdots X_n\} \setminus \{X_p\}$, we have $\mathcal{T}_{(X_p|\mathbf{S})}$ is full rank. The reason is the following.

Since $\mathbb{P}(X_p|\mathbf{S})\mathbb{P}(\mathbf{S}) = \mathbb{P}(\mathbf{S}|X_p)\mathbb{P}(X_p)$ by Bayes' theorem Koch and Koch (1990), and due to all marginal probabilities are not zero, then we have $\mathcal{T}_{(X_p|\mathbf{S})} = \mathcal{T}_{(\mathbf{S}|X_p)}\text{Diag}(\mathcal{T}_{(X_p)})\text{Diag}(\mathcal{T}_{(\mathbf{S})})^{\dagger}$, where $\dagger$ is inverse of matrix, and $X_p$ is ancestor of $\mathbf{S}$. By Lemma .D.2, $\mathbf{S}$ has a parent variables and hence $\mathcal{T}_{(\mathbf{S}|X_p)}$ is full rank (one can vectorize the variable set $\mathbf{S}$ as a variable). Now, we have $\mathcal{T}_{(X_p|\mathbf{S})}$ is full rank due to the three matrices on the right are all full rank.

Thus, $\mathcal{T}_{(X_p|\mathbf{S}=1)} = \alpha_2\mathcal{T}_{(X_p|\mathbf{S}=2)} = \cdots\mathcal{T}_{(X_p|\mathbf{S}=r)}$ does not hold, i.e., $\mathcal{T}_{(X_1\cdots X_n|\tilde{\mathbf{S}}=j)}$ is not a rank-one tensor.

$\square$

**Lemma D.4.** *In a discrete causal graph, for the variable $X_p$, let $L_p$ represent the vectorized parent set of $X_p$, and let $\text{Des}_{X_p}$ denote the set of descendant variables of $X_p$. Under this setup, $\mathcal{T}_{(X_p|L_p,\text{Des}_{X_p}=j)}$ is full rank.*

*Proof.* By Bayes' theorem Koch and Koch (1990), we have

$$
\begin{aligned}
&\mathcal{T}_{(X_p, L_p | \mathrm{Des}_{X_p} = j)} \\
&= \mathcal{T}_{(X_p | L_p, \mathrm{Des}_{X_p} = j)} \mathrm{Diag}(\mathcal{T}_{(L_p | Des_{X_p} = j)}) \\
&= \mathcal{T}_{(L_p | X_p, \mathrm{Des}_{X_p} = j)} \mathrm{Diag}(\mathcal{T}_{(X_p | \mathrm{Des}_{X_p} = j)}).
\end{aligned}
\tag{16}
$$

Since $X_p$ and $\mathrm{Des}_{X_p}$ both are the descendant set of $L_p$, we have

$$
\mathcal{T}_{(L_p | \mathrm{Des}_{L_p})} = \mathcal{T}_{(\mathrm{Des}_{L_p} | L_p)} \mathrm{Diag}(\mathcal{T}_{(\mathrm{L}_p)} \mathrm{Diag}(\mathcal{T}_{(\mathrm{Des}_{L_p})})^{\dagger},
\tag{17}
$$

due to the fact that all marginal distribution probabilities are not zero. Then $\mathcal{T}_{(L_p | \mathrm{Des}_{L_p})}$ is full rank, i.e., $\mathcal{T}_{(L_p | X_p, \mathrm{Des}_{X_p} = j)}$ also full rank.

Moreover, for any $\mathcal{T}_{(L_p = i | \mathrm{Des} = j)}$, we have

$$
\mathcal{T}_{(L_p = i | \mathrm{Des}_{X_p} = j)} = \frac{\mathcal{T}_{(L_p = i, \mathrm{Des}_{X_p} = j)}}{\mathcal{T}_{(\mathrm{Des}_{X_p} = j)}} \neq 0,
\tag{18}
$$

because all marginal distribution probabilities are not zero. Thus, we have

$$
\begin{aligned}
&\mathcal{T}_{(X_p, L_p | \mathrm{Des}_{X_p} = j)} \\
&= \mathcal{T}_{(L_p | X_p, \mathrm{Des}_{X_p} = j)} \mathrm{Diag}(\mathcal{T}_{(X_p | \mathrm{Des}_{X_p} = j)}) \mathrm{Diag}(\mathcal{T}_{(L_p | \mathrm{Des}_{X_p} = j)})^{\dagger}.
\end{aligned}
\tag{19}
$$

One can see that $\mathcal{T}_{(X_p, L_p | \mathrm{Des}_{X_p} = j)}$ is full rank due to three matrices on the right side being full rank.

$\square$

### D.2 Proof of Proposition 4.2

*Proof.* The proof is straightforward. In the discrete 3PLSM model, suppose all latent variable has the same state space. Any two observed are d-separated by any one of their latent parents. According to the graphical criteria, the rank of tensor $\mathcal{T}_{(X_i X_j)}$ is the dimension of latent support.

$\square$

### D.3 Proof of Proposition 4.3

*Proof.* **Proof of $\mathcal{R}$ule 1:** In the discrete 3PLSM and suppose the assumption 2.2 ~ assumption 2.4 holds, if there does not exist a latent variable $L_1$ that d-separates any pair variables in $\{X_i, X_j, X_k\}$, i.e., the rank of tensor $\mathcal{T}_{(X_i X_j X_k)}$ is not $r$ (by Theorem 1), it must be the full-connection structure among latent variables, as shown in Fig. 4 (d). Otherwise, one can find one latent variable $L_1$ that can d-separates $\{X_i, X_j, X_k\}$, as shwon in Fig. 4 (a) ~ (c). We will show that, if only consider one of the latent variables of $\{L_1, L_2, L_3\}$ in Fig. 4 (d), the tensor of $\mathcal{T}_{(X_i X_j X_k)}$ can not have rank-one decomposition.

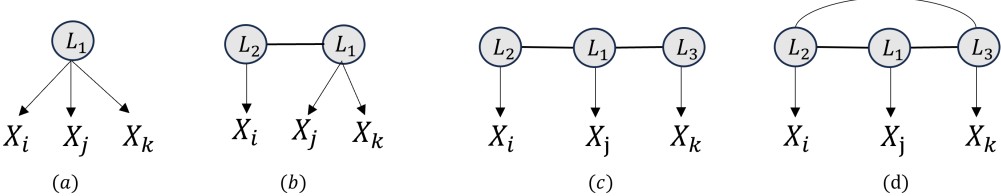

Figure 4: Illustrative example for $\mathcal{R}$ule 1.

According to the graphical criteria of tensor rank and $r < d_i$ in the definition of discrete LSM, and suppose all latent variables have the same state space, for $X_i$, $X_j$ and $X_k$, there is not only one latent

variable is conditional set, i.e., but there also is not one latent variable $L$ that d-separates any pair variables in $\{X_i, X_j, X_k\}$. Thus, $\mathrm{Rank}(\mathcal{T}_{(X_i X_j X_k)}) \neq r$.

**Proof of $\mathcal{R}$ule 2:**

We aim to show if $\mathrm{Rank}(\mathcal{T}_{(X_i X_j X_k X_s)}) = r$ for any $X_s \in \mathbf{X} \setminus \{X_i, X_j, X_k\}$ then there exists a latent variable $L_p$ that d-separates $\{X_i, ..., X_s\}$ and $L_p$ is the parent variable of $\{X_i, X_j, X_k\}$ in the discrete 3PLSM. We first prove it by the contradiction. If $\{X_i, X_j, X_k\}$ does not share one common latent parent, e.g., $L_1 \to \{X_i, X_j\}$ and $L_2 \to \{X_k\}$, due to the structure assumption in discrete 3PLSM, there exist $X_s \in Ch_{L_2}$, as shown in Fig. 5.

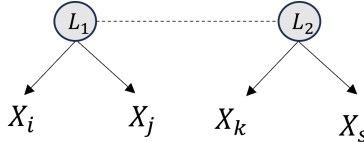

Figure 5: Illustrative example for $\mathcal{R}$ule 2.

By the graphical criteria of tensor rank condition and $r < d_i$, one has $\mathrm{Rank}(\mathcal{T}_{(X_i X_j X_k X_s)}) \neq r$ since $L_1$ or $L_2$ is not the conditional set that d-separates any pair variable of $\{X_i, X_j, X_k, X_s\}$ (e.g., $X_k, X_s$ can not be d-separates given $L_1$), which is contrary to the condition $\mathrm{Rank}(\mathcal{T}_{(X_i X_j X_k X_s)}) = r$. $\qquad\square$

## D.4 Proof of Proposition 4.5

*Proof.* Since $C_1$ and $C_2$ are two causal clusters, then the elements in $C_1$ have only one common latent variable. Without loss of generality, we let $L_1$ denote the parental latent variable of $C_1$. Similarly, $L_2$ denotes the parental latent variable of $C_2$. Since $C_1$ and $C_2$ are overlapping, then they have at least one shared element. Let $X_i$ denote the shared element of $C_1$. then $X_i$ has two latent parents $L_1$ and $L_2$, which contradicts with the pure child assumption in the discrete 3PLSM model. This finishes the proof. $\qquad\square$

## D.5 Proof of Theorem 4.6

*Proof.* Based on Prop. 4.3, one can identify the causal cluster by testing the tensor rank condition (Line 5 ~ 12). Moreover, the Prop. 4.5 ensures no redundant latent variables are introduced (Line 15). Thus, the causal cluster can be identified by Algorithm 1, under the discrete 3PLSM, with assumption 2.2 ~ assumption 2.4. $\qquad\square$

## D.6 Proof of Theorem 4.7

*Proof.* We first prove this result by a specific case and then extend it to a general case result.

**Proof by Specific case**

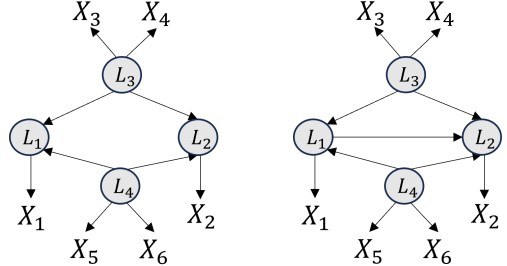

Figure 6: d-separation and d-connection example.

'If' part: as shown in Fig.6, suppose all support of latent space is $r$ and $r < d_i$, $d_i$ is the dimension of any support of observed variables. We will show that, for $\{X_1, ..., X_6\}$, the rank of $\mathcal{T}_{\{X_1,...,X_6\}}$ are $r^2$. Let $\mathbf{L}$ be the vectorization of joint distribution $\mathbb{P}(L_3, L_4)$ with $|\text{supp}(\mathbf{L})| = r^2$.

Since $L_2, L_3$ is a conditional set that d-separates all variables in $\{X_1, \cdots, X_6\}$ and there is no other conditional set with smaller support, according to the graphical criteria of tensor rank, $\text{Rank}(\mathcal{T}_{(X_1 \cdots X_6)}) = |\text{supp}(\mathbf{L})| = r^2$.

'Only if' part: now, we consider the case that if $L_1$ and $L_2$ are d-connection (Fig.6). In this case, given $\mathbb{P}(L_3 L_4)$ (represented by $\mathbb{P}(\mathbf{L})$), $\mathcal{T}_{(X_1 X_2 | \mathbf{L}=i)}$ cannot be decomposition as the outer product of two vectors since the contingency table of $\mathcal{T}_{(X_1 X_2 | \mathbf{L}=i)}$ is not a rank-one tensor by Markov assumption. That is, $\mathbf{L}$ is not a conditional set that d-separates $X_1$ and $X_2$.

According to the graphical criteria of tensor rank condition, one can infer that $\text{Rank}(\mathcal{T}_{(X_1 \cdots X_6)}) \neq |\text{supp}(\mathbf{L})| = r^2$. Based on the above analysis, one can see that the conditional independent relations hold if and only if the rank of the tensor equals the cardinality of support of the conditional set.

**Proof by general case**

'if' part: in the discrete 3PLSM model, assume that all latent variables have the same support. For any pair of variables $X_i$ and $X_j$ that are pure children of $L_k$, it is evident that $L_k$ is the only minimal conditional set that d-separates $X_i$ from $X_j$. Consider a set of latent variables $\mathbf{L}_p$ and their corresponding child sets $\mathbf{X}_{p1}$ and $\mathbf{X}_{p2}$, where each latent variable has at least two child variables included in $\mathbf{X}_{p1}$ and $\mathbf{X}_{p2}$. The minimal conditional set that d-separates all variables in $\mathbf{X}_{p1}$ from those in $\mathbf{X}_{p2}$ is their common latent parent set $\mathbf{L}_p$. Moreover, for two variables $X_i$ and $X_j$ that do not share a common latent parent, if $\mathbf{L}_p$ d-separates $X_i$ and $X_j$, then $\mathbf{L}_p$ also d-separates all variables in $\mathbf{X}_{p1} \cup \mathbf{X}_{p2} \cup \{X_i, X_j\}$. Since all latent variables have the same support, the minimal conditional set in the graph corresponds to the set with the minimal support. Therefore, by the graphical criteria of tensor rank, $\text{Rank}(\mathcal{T}(X_i, X_j, \mathbf{X}_{p1}, \mathbf{X}_{p2})) = |\text{supp}(\mathbf{L}_p)|$. As all latent variables share the same state space, we deduce that $|\text{supp}(\mathbf{L}_p)| = r^{|\mathbf{L}_p|}$.

'Only if' part: if $X_i$ and $X_j$ cannot be d-separated by $\mathbf{L}_p$, then $\mathbf{L}_p$ does not constitute a conditional set for $\mathbf{X}_{p1} \cup \mathbf{X}_{p2} \cup \{X_i, X_j\}$, under Assumptions 2.2 ~ 2.4. According to the graphical criteria of tensor rank, $\text{Rank}(\mathbb{P}(X_i, X_j, \mathbf{X}_{p1}, \mathbf{X}_{p2})) \neq |\text{supp}(\mathbf{L}_p)|$.

$\square$

### D.7 Proof of Theorem 4.9

*Proof.* Such an identification is derived from the original PC algorithm. By Theorem 4.7, given the measurement model, one can test the CI relations among latent variables, when $r < d_i$ (remark 1). Thus, the causal structure among latent variables can be identified up to a Markov equivalent class by Algorithm 2 Spirtes et al. (2000). $\square$

## E   Extension of Different Latent State Space

To extend our theoretical result to the case in which the state space of the latent variable may be different (i.e., $r_i \neq r_j$). We present the minimal state space criteria by which the state space of the latent variable in the conditional set is identifiable.

**Theorem E.1** (Minimal state space criteria). *In the discrete 3PLSM, suppose Assumption 2.2 ~ 2.4 holds. For any two observed variables $X_i$ and $X_j$, let $L_i$ and $L_j$ be their latent parent respectively, and let $r_1$ be the cardinality of $\text{supp}(L_i)$ and $r_2$ be the cardinality of $\text{supp}(L_j)$, there is $\text{Rank}(\mathcal{T}_{(X_1, X_2)}) = min(r_1, r_2)$.*

*Proof.* In the discrete 3PLSM model, for two observed variables, we have $r < x_i$ and any one of the latent parents can d-separate $X_i$ from $X_i$. Based on the graphical criteria of the tensor rank condition, one can see that the rank is the latent parent with minimal cardinality of support. $\square$

Based on the minimal state space criteria, one can directly extend the identification of the discrete 3PLSM to the case where the cardinality of latent support may be different.

## E.1 Identification of causal cluster

**Proposition E.2** (Identification of causal cluster in different state space). *In the discrete 3PLSM, suppose Assumption 2.2 ~ Assumption 2.4 holds. For three disjoint observed variables $X_i, X_j, X_k \in \mathbf{X}$, then $\{X_i, X_j, X_k\}$ share the same latent parent if $\mathrm{Rank}(\mathcal{T}_{(X_i, X_j, X_k, X_s)}) = r$ for any $X_s \in \mathbf{X} \setminus \{X_i, X_j, X_k\}$, where $r = \mathrm{Rank}(\mathcal{T}_{(X_i, X_j)}) = \mathrm{Rank}(\mathcal{T}_{(X_i, X_k)}) = \mathrm{Rank}(\mathcal{T}_{(X_k, X_j)})$.*

*Proof.* If $\mathrm{Rank}(\mathcal{T}_{(X_i, X_j, X_k, X_s)}) = r$, there exist a variable set $\mathbf{S}$ with $|\mathrm{supp}(\mathbf{S})| = r$ that d-separates any pair variables in $\{X_i, X_j, X_k, X_s\}$, according to the graphical criteria of tensor rank condition. In the discret LSM model and $r < d_i$ (any support dimension of latent variable is less than the support dimension of observed variable), $r = \mathrm{Rank}(\mathcal{T}_{(X_i, X_j)}) = \mathrm{Rank}(\mathcal{T}_{(X_i, X_k)}) = \mathrm{Rank}(\mathcal{T}_{(X_k, X_j)})$, we have for any $X_i, X_j \in \{X_i, X_j, X_k, X_s\}$, the conditional set is one of the latent parent of $X_i$ and $X_j$. If $X_i, X_j$ and $X_k$ do not share the common latent parent, without loss of generality, let $L_1$ be the parent of $X_i, X_j$ and $L_2$ be the parent of $X_k$, there exist the $X_s \in Ch_{L_2}$ such that $X_k$ and $X_s$ cannot be d-separated given $L_1$ or $X_i$ and $X_j$ cannot be d-separated given $L_2$. By the graphical criteria of tensor rank condition, $\mathrm{Rank}(\mathcal{T}_{(X_i, X_j, X_k, X_s)}) \neq |\mathrm{supp}(L_1)|$, or $\mathrm{Rank}(\mathcal{T}_{(X_i, X_j, X_k, X_s)}) \neq |\mathrm{supp}(L_2)|$. Let $r = \min(|\mathrm{supp}(L_1)|, |\mathrm{supp}(L_2)|)$, we have $\mathrm{Rank}(\mathcal{T}_{(X_i, X_j, X_k, X_s)}) \neq r$ if $X_i, X_j$ and $X_j$ are not a causal cluster. $\square$

One can properly adjust the search algorithm such that the causal cluster can be identified, by the minimal state space criteria. The algorithm is presented as follows (Algorithm 3).

---

**Algorithm 3** Identifying the causal cluster (different latent space)

---

**Input**: Data from a set of measured variables $\mathbf{X}_\mathcal{G}$
**Output**: Causal cluster $\mathcal{C}$

1: Initialize the causal cluster set $\mathcal{C} := \varnothing$, and $\mathcal{G}' = \varnothing$;
2: **Begin** the recursive procedure
3: **repeat**
4:    **for** each $X_i, X_j$ and $X_k \in \mathbf{X}$ **do**
5:       *// Apply the minimal state space criteria*
6:       r = $\min(\mathrm{Rank}(\mathcal{T}_{\{X_i, X_j\}}), \mathrm{Rank}(\mathcal{T}_{\{X_i, X_k\}}), \mathrm{Rank}(\mathcal{T}_{\{X_k, X_j\}}))$;
7:       **if** $\mathrm{Rank}(\mathcal{T}_{\{X_i, X_j, X_k, X_s\}}) = r$, for all $X_s \in \mathbf{X} \setminus \{X_i, X_j, X_k\}$ **then**
8:          $\mathbf{C} = \mathbf{C} \cup \{\{X_i, X_j, X_k\}\}$;
9:       **end if**
10:    **end for**
11: **until** no causal cluster is found.
12: *// Merging cluster and introducing latent variables*
13: Merge all the overlapping sets in $\mathbf{C}$ by Prop. 4.5.
14: **for** each $C_i \in \mathbf{C}$ **do**
15:    Introduce a latent variable $L_i$ for $C_i$;
16:    $\mathcal{G} = \mathcal{G} \cup \{L_i \to X_j | X_j \in C_i\}$.
17: **end for**
18: **return** Graph $\mathcal{G}$ and causal cluster $\mathcal{C}$.

---

## E.2 Conditional independence test among latent variables

**Proposition E.3** (conditional independence among latent variables in different state space). *In the discrete 3PLSM, suppose Assumption 2.2 ~ Assumption 2.4 holds. Let $X_i$ and $X_j$ be the pure child of $L_i$ and $L_j$ respectively, $\mathbf{X}_{p1}$ and $\mathbf{X}_{p2}$ be two disjoint child set of the latent set $\mathbf{L}_p$ with $|\mathbf{X}_{p1}| = |\mathbf{X}_{p2}| = |\mathbf{L}_p|$, then $L_i \perp L_j | \mathbf{L}_p$ if and only if $\mathrm{Rank}(\mathcal{T}_{(X_i, X_j, \mathbf{X}_{p1}, \mathbf{X}_{p2})}) = r$, where $r = \prod_{L_i \in \mathbf{L}_p} |\mathrm{supp}(L_i)|$.*

*Proof.* 'If' part: in the discrete 3PLSM, we have $r_i < d_j$ for any $i \in [k], j \in [p]$, where $k$ is the number of latent variables while $p$ is the number of observed variables. In the causal graph, for $X_{q1} \in \mathbf{X}_{p1}$ and $X_{q2} \in \mathbf{X}_{p2}$, $X_{q1}, X_{q2} \in Ch(L_t)$ for $\forall L_t \in \mathbf{L}_p$, we have $L_t$ is the only conditional set that d-separates $X_{q1}$ and $X_{q2}$ with minimal support dimension. Thus, $\mathbf{L}_p$ also be

the minimal conditional set that d-separates any pair variables in $\mathbf{X}_{p1} \cup \mathbf{X}_{p2}$. Now, if $X_i$ and $X_j$ are d-separated by $\mathbf{L}_p$, according to the graphical criteria of tensor rank condition, one have $\mathrm{Rank}(\mathcal{T}_{(X_i,X_j,\mathbf{X}_{p1},\mathbf{X}_{p2})}) = |\mathrm{supp}(\mathbf{L}_p)|$. Since $\mathbf{L}_p$ is the joint distribution of latent variable set, we have $r = \prod_{L_i \in \mathbf{L}_p} |\mathrm{supp}(L_i)|$.

'Only if' part: on the other hand, if $X_i$ and $X_j$ are not d-separated by $\mathbf{L}_p$, for example, $L_i \to L_j$ in the causal graph, then $\mathbf{L}_p$ is not a conditional set that d-separates all pair variables in $\{X_i, X_j, \mathbf{X}_{p1}, \mathbf{X}_{p2}\}$. According to the graphical criteria of tensor rank condition, we have $\mathrm{Rank}(\mathcal{T}_{(X_i,X_j,\mathbf{X}_{p1},\mathbf{X}_{p2})}) \neq |\mathrm{supp}(\mathbf{L}_p)|$, i.e., $\mathrm{Rank}(\mathcal{T}_{(X_i,X_j,\mathbf{X}_{p1},\mathbf{X}_{p2})}) \neq \prod_{L_i \in \mathbf{L}_p} |\mathrm{supp}(L_i)|$. This completes the proof. $\square$

In particular, $|\mathrm{supp}(L_i)|$ can be identified by their pure child variable, according to minimal state space criteria. An intuition illustration is by mapping the conditional set variable $\mathbf{L}_p$ to one new latent variable $\tilde{L}$ (i.e., vectorization), the graphical criteria of causal cluster still hold, e.g., $\{\mathbf{X}_{p1}, \mathbf{X}_{p2}, X_i\}$ is a causal cluster that shares a common parent $\tilde{L}$. However, such a map will exponentially increase the dimension of the latent variable support. One issue will be raised: the observed variable may have a smaller support dimension than the latent variables such that the d-separation relations among latent variables cannot be examined. Thus, it is necessary to study when and how the testability of d-separation holds. The result is provided in Remark. E.4.

**Remark E.4** (Testability of d-separation). *For an n-way tensor $\mathcal{T}_{\{X_1,\dots,X_n\}}$ that is used to test the d-separation relations among latent variables, such a CI relation is testable if $\prod_{j=1}^{|\mathbf{L}_p|} r_j^{|\mathbf{L}_p|} < \prod_{i=1}^{n} d_i - max(d_1, \dots, d_n)$.*

*Proof.* We prove it by contradiction. If $\prod_{j=1}^{|\mathbf{L}_p|} r_j^{|\mathbf{L}_p|} > \prod_{i=1}^{n} d_i - \max(d_1, \dots, d_n)$, assume that $d_n = max(d_1, \cdots, d_n)$ where $d_i$ is the support of observed $X_i$, there are

$$
\begin{aligned}
\mathcal{T}_{(X_1 \cdots X_n)} &= \sum_{\mathbb{P}(X_1 \cdots X_{n-1})} \mathbb{P}(X_1 | X_1 \cdots X_{n-1}) \cdots \mathbb{P}(X_n | X_1 \cdots X_{n-1}) \mathbb{P}(X_1 \cdots X_{n-1}) \\
&= \sum_{i=1}^{\prod_{j=1}^{n-1} d_j} \mathbf{u}_1^{(i)} \otimes \cdots \otimes \mathbf{u}_1^{(i)},
\end{aligned}
\tag{20}
$$

which is a smaller rank-one decomposition than $\mathbf{L}_p$ with support $r^{|\mathbf{L}_p|}$. According to the definition of tensor rank and the graphical criteria of tensor rank, we have $\mathrm{Rank}(\mathcal{T}_{(X_1 \cdots X_n)}) = \prod_{i=1}^{n} d_i - \max(d_1, \dots, d_n)$. That is, no matter whether the conditional independent relations hold given $\mathbf{L}_p$, the rank of tensor still be the dimension of the support of $\mathbb{P}(X_1 \cdots X_{n-1})$. It means that the CI relations can not be detected.

$\square$

In other words, if the support of the conditional set is more than the dimension of observed variables, then the minimal rank-one decomposition of the joint distribution will lead to $\mathbb{P}(X_1, \cdots, X_n) = \sum_{\{X_1, \cdots, X_{n-1}\}} \mathbb{P}(X_1, \cdots, X_n | X_1, \dots, X_{n-1}) P(X_1, \cdots, X_{n-1})$. Thus, if the increasing of latent state space is less than the sum of tensor dimensions, the CI relations among latent variable are testable. Due to assuming that $d > r$, the CI relations among latent variables are generally testable when the causal structure is sparse.

## F    Discussion with the Hierarchical Structures

Actually, our result can be extended to a specific hierarchical structure, by constraining the structure of hidden variables. For instance, consider a hierarchical structure in which each latent variable is required to have at least three pure children (whether latent or observed) and one additional neighboring variable. An illustration of this type of structure is provided in Fig. 7. Assume that all latent variables have the same dimension of support, and that this dimension is smaller than that of the observed variables. Under these conditions, causal clusters at the bottom level can still be identified, as demonstrated by Proposition 2. For instance, the sets $\{X_1, X_2, X_3\}$,

$\{X_4, X_5, X_6\}$, $\{X_7, X_8, X_9\}$, and $\{X_{10}, X_{11}, X_{12}\}$ are recognized as four distinct causal clusters. The pure measured variables from each cluster can act as surrogates for their corresponding latent parents, allowing the causal cluster learning procedure to be repeated. For example, if $X_1$ serves as the surrogate for $L_2$, and $X_4, X_7, X_{10}$ as surrogates for $L_3, L_4, L_5$, then $\{L_2, L_3, L_4, L_5\}$ can be identified as a cluster according to the graphical criteria of tensor rank. Thus, the specific hierarchical structure is identifiable by designing the proper search algorithm making use of the tensor rank condition. We will explore these results in future works.

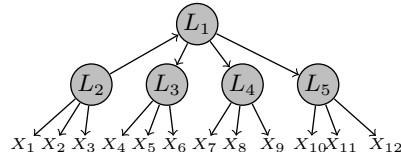

Figure 7: Example of hierarchical structure.

# G  Practical Estimation of Tensor Rank

Here, we describe the practical implementation of tensor rank estimation. To alleviate the problem of local optima during tensor decomposition, we initiate the process from multiple random starting points. We then perform the tensor decomposition from each of these points and subsequently select the decomposition that yields the smallest reconstruction error as our final result. The procedure is summarised in the Algorithm 4.

---
**Algorithm 4** Practial tensor rank estimation
---
**Input**: $\mathbf{X}_p = \{X_i, X_j, ..., X_n\}$, iteration number $n$, threshold $\varepsilon_r$ and tested rank $r$
**Output**: Boolean of rank test
1: Initialize the minimal reconstructed error $E_{min} = +\infty$;
2: **for** $i < n$ or $E_{min} \leq \varepsilon$ **do**
3:    $i \leftarrow i + 1$;
4:    $\tilde{\mathcal{T}} \leftarrow$ non-negative-parafac($\mathcal{T}_{(\mathbf{X}_p)}, r$);
5:    E $\leftarrow \|\mathcal{T}_{(\mathbf{X}_p)} - \tilde{\mathcal{T}}\|$;
6:    **if** $E \leq E_{min}$ **then**
7:       $E_{min} = E$;
8:    **end if**
9: **end for**
10: p-val $\leftarrow$ Chi-Square($\mathcal{X}^2$)-Test(vec($\mathcal{T}$), vec($\tilde{\mathcal{T}}$));
11: **if** p-val $< \varepsilon$ **then**
12:    **return True**;
13: **end if**
14: **return False**.
---

Besides, in the PC-TENSOR-RANK algorithm, to further identify the V-structure among latent variables, the statistic-independent test among latent variables is required, which can be tested by following.

**Remark G.1** (Statistic independent between latent variables). *Give the measured variable $X_i$ and $X_j$ of latent variable $L_i$ and $L_j$, then $L_i \perp\!\!\!\perp L_j$ if Rank$(\mathbb{P}(X_i, X_j)) = 1$.*

*Proof.* Since $L_i \perp\!\!\!\perp L_j$, we have $X_i \perp\!\!\!\perp X_j$ also hold in the causal graph. We have $\mathbb{P}(X_i X_j) = \mathbb{P}(X_i)\mathbb{P}(X_j) = \mathcal{T}_{(X_i)} \otimes \mathcal{T}_{(X_j)}$. According to the definition of tensor rank, Rank$(\mathcal{T}_{(X_i, X_j)}) = 1$. $\quad\square$

## G.1  Goodness of fit test for CI test among latent variables

Although the proposed tool is theoretically testable, it still is an approximate estimation of tensor rank by heuristic-based CP decomposition in practice. How to consider a more robust approach to examine the tensor rank still be an open problem in the related literature. It significantly restricts

the application scope and performance of our structure learning algorithm. However, we want to emphasize that the main contribution of our work is building the graphical criteria of tensor rank and using it to answer the identification of causal structure in a discrete 3PLSM model.

Next, we will show how the CI relations among latent variables can be distinguished by testing the goodness of fit test. Consider a four latent variables structure as shown in Fig. 8, a chain structure among four latent variables in which each latent variable has two pure observed variables. The data generation process follows the discrete 3PLSM model (see the description in the simulation studies section) and the sample size is 50k. We check the CI relations between any $L_i, L_j \in \{L_1, L_2, L_3, L_4\}$ given $L_p \in \{L_1, L_2, L_3, L_4\} \setminus \{L_i, L_j\}$. The results are reported on the right side of Fig. 8, in which each red point represents a CI test result, e.g., the second point in the graph represents to test $L_2 \perp\!\!\!\perp L_4 | L_1$ by examining $\mathrm{Rank}(\mathcal{T}_{(X_3, X_7, X_1, X_2)}) = 2$. One can see that the p-value returned by the goodness of fit test is lower than 0.05, which means that we will tend to reject the null hypothesis, i.e., $\mathrm{Rank}(\mathcal{T}_{(X_3, X_7, X_1, X_2)}) \neq 2$. By sorting all CI test results, one can see that the true CI relations can be identified by setting the significant level to be 0.05.

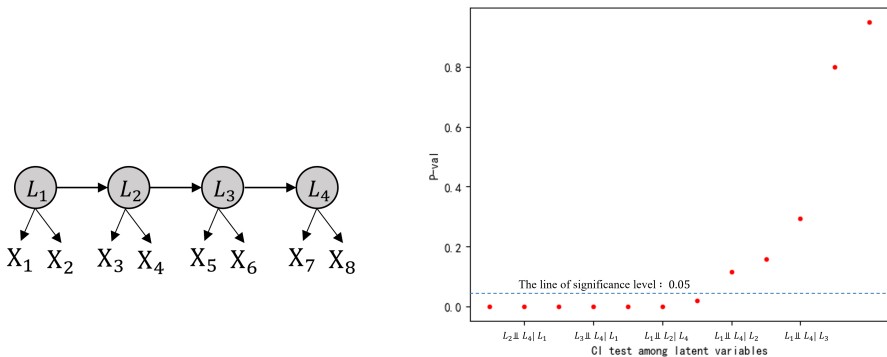

Figure 8: Goodness of fit test for conditional independent test among latent variables

# H   More Experimental Results

In this section, we provide the information required to reproduce our results reported in the main text. We further conduct additional simulation experiments to validate the efficiency of the proposed algorithm (Appendix E).

We first give a details definition of evaluation metrics. Specifically, the performance of causal cluster is evaluated by following scores for the output model $G_{out}$ from each algorithm, where the true graph is labelled $G$:

- **latent omission**, the number of latents in $G$ that do not appear in $G_{out}$ divided by the total number of true latents in $G$;

- **latent commission**, the number of latents in $G_{out}$ that could not be mapped to a latent in $G$ divided by the total number of true latents in $G$;

- **mismeasurement**, the number of observed variables in $G_{out}$ that are measuring at least one wrong latent divided by the number of observed variables in $G$;

Moreover, we use the following metric to evaluate the performance of causal structure among latent variables:

- **edge omission (EO)**, the number of edges in the structural model of $G$ that do not appear in $G_{out}$ divided by the possible number of edge omissions;

- **edge commission (EC)**, the number of edges in the structural model of $G_{out}$ that do not exist in $G$ divided by the possible number of edge commissions;

- **orientation omission (OO)**, the number of arrows in the structural model of $G$ that do not appear in $G_{out}$ divided by the possible number of orientation omissions in $G$;

These evaluation indicators are derived from Silva et al. (2006).

Next, we give a concrete implementation of baseline methods.

**BayPy**: The Bayesian Pyramid Mode (BayPy) is a discrete latent variable structure learning method that assumes the latent structure is a pyramid structure and the latent variable is binary. We use the implementation of Gu and Dunson (2023). We set the iteration parameter to 1500 and set the search upper bound of the number of latent variables to 5.

**LTM**: The latent tree model, is a classic method for learning gaussian or binary latent tree structure. We use the implementation from Choi et al. (2011). Specifically, we use the Recursive Grouping (RG) Algorithm in Choi et al. (2011) (since it has better performance), and use the discrete information distance to learn the structure of the discrete 3PLSM model.

**BPC**: The Building Pure Cluster (BPC) algorithm Silva et al. (2006) is a classic causal discovery method for the linear latent variable model. We use the implementation from the Tetrad Project package [3].

**Non-negative CP decomposition**: To perform non-negative CP decomposition in our algorithm, we use the implementation from the python package, *tensorly* [4], and set the maximum iteration parameter to 1000, the cvg criterion parameter to "rec_error".

**Data Generation**: To generate the probability contingency table or conditional probability contingency table for latent variables and observed variables in our simulation studies, we use the implementation from the python package, *pgmpy* [5]. The package provides the function to generate observed data according to the probability contingency table.

Finally, we aim to demonstrate the correctness of our methods in handling cases involving latent variables with varying state spaces. Specifically, we consider the structure model with star structure and the $SM_3$ structure, and the measurement model with $MM_1$. The data generation process follows the description in the main context and we let the support of latent variable $L_1, L_2$ to be $\{0, 1\}$ and the support of latent variable $L_3, L_4$ to be $\{0, 1, 2\}$, for simulating the case that latent variable has different latent support. Besides, the support of the observed variable is $\{0, 1, 2, 3\}$. In our implementation, the significant levels for testing the rank of the matrix and tensor are set to 0.005 and 0.05, respectively. The coefficient of probability contingency tables is generated randomly, ranging from $[0.1, 0.8]$, constraining the sum of them to be one.

The results are presented in Table 4 and Table 5. The performance of our method appears superior in scenarios where latent variables have differing state spaces. This improvement is attributed to the reduction in the frequency of higher-order tensor rank testing, facilitated by evaluating the consistency of ranks such as $\mathrm{Rank}(\mathcal{T}(X_i, X_j)) = \mathrm{Rank}(\mathcal{T}(X_i, X_k)) = \mathrm{Rank}(\mathcal{T}_{(X_k, X_j)})$. In contrast, the BayPy approach underperforms in latent structure learning, likely due to its assumption of a pyramid structure with a top-down directionality and no peer-level connections among latent variables. Additionally, the Latent Tree Model (LTM) shows weaker performance in cluster learning, possibly because it was originally designed to handle only binary discrete variables.

Table 4: Results on learning pure measurement models in the case that latent variables have different state spaces. Lower value means higher accuracy.

| Algorithm | | Latent omission | | | | Latent commission | | | | Mismeasurements | | | |
|---|---|---|---|---|---|---|---|---|---|---|---|---|---|
| | | **Our** | BayPy | LTM | BPC | **Our** | BayPy | LTM | BPC | **Our** | BayPy | LTM | BPC |
| | 5k | 0.09(3) | 0.20(4) | 0.23(5) | 0.96(10) | 0.00(0) | 0.20(4) | 0.00(0) | 0.00(0) | 0.03(1) | 0.18(4) | 0.23(5) | 0.00(0) |
| $Star + MM_1$ | 10k | 0.06(2) | 0.17(3) | 0.13(4) | 0.96(10) | 0.00(0) | 0.15(3) | 0.00(0) | 0.00(0) | 0.00(0) | 0.15(3) | 0.13(4) | 0.00(0) |
| | 50k | 0.00(0) | 0.13(3) | 0.10(3) | 0.93(10) | 0.00(0) | 0.15(3) | 0.00(0) | 0.00(0) | 0.00(0) | 0.13(3) | 0.10(3) | 0.00(0) |
| | 5k | 0.12(3) | 0.33(7) | 0.55(10) | 0.96(10) | 0.00(0) | 0.30(6) | 0.00(0) | 0.00(0) | 0.06(2) | 0.30(7) | 0.55(10) | 0.00(0) |
| $SM_3 + MM_1$ | 10k | 0.06(2) | 0.26(5) | 0.50(10) | 0.93(10) | 0.00(0) | 0.20(5) | 0.00(0) | 0.00(0) | 0.00(0) | 0.19(5) | 0.50(10) | 0.00(0) |
| | 50k | 0.03(1) | 0.20(4) | 0.50(10) | 0.93(10) | 0.00(0) | 0.15(4) | 0.00(0) | 0.00(0) | 0.00(0) | 0.11(4) | 0.50(10) | 0.00(0) |

---

[3] https://github.com/cmu-phil/tetrad

[4] https://github.com/tensorly/tensorly

[5] https://github.com/pgmpy/pgmpy

Table 5: Results on learning the structure model in the case that latent variables have different state spaces. The symbol '-' indicates that the current method does not output this information. Lower value means higher accuracy.

| Algorithm | | Edge omission | | | | Edge commission | | | | Orientation omission | | | |
|---|---|---|---|---|---|---|---|---|---|---|---|---|---|
| | | **Our** | BayPy | LTM | BPC | **Our** | BayPy | LTM | BPC | **Our** | BayPy | LTM | BPC |
| $Star+MM_1$ | 5k | 0.00(0) | 1.00(10) | 0.26(8) | 1.00(10) | 0.10(1) | 0.00(0) | 0.00(0) | 0.00(0) | 0.10(1) | 1.00(10) | – | 1.00(0) |
| | 10k | 0.00(0) | 1.00(10) | 0.23(6) | 1.00(10) | 0.00(0) | 0.02(1) | 0.0(0) | 0.00(0) | 0.00(0) | 1.00(10) | – | 1.00(0) |
| | 50k | 0.00(0) | 1.00(10) | 0.10(3) | 1.00(10) | 0.00(0) | 0.00(0) | 0.00(0) | 0.00(0) | 0.00(0) | 1.00(10) | – | 1.00(0) |
| $SM_3 + MM_1$ | 5k | 0.15(3) | 1.00(10) | 0.16(6) | 1.00(10) | 0.10(1) | 0.00(0) | 0.00(0) | 0.00(0) | 0.00(0) | 0.00(0) | – | 0.00(0) |
| | 10k | 0.05(1) | 1.00(10) | 0.13(4) | 1.00(10) | 0.01(1) | 0.00(0) | 0.00(0) | 0.00(0) | 0.00(0) | 0.00(0) | – | 0.00(0) |
| | 50k | 0.00(0) | 1.00(10) | 0.10(3) | 1.00(10) | 0.00(0) | 0.00(0) | 0.00(0) | 0.00(0) | 0.00(0) | 0.00(0) | – | 0.00(0) |

# I   Experimental Results on Real-world Dataset

We first briefly present the results from two real datasets. The first is the political efficacy dataset, collected by Aish and Jöreskog (1990) through a cross-national survey designed to capture information on both conventional and unconventional forms of political participation in industrial societies. This dataset includes 1719 cases obtained in a USA sample. The second dataset, referred to as the depress dataset, is detailed by Jöreskog and Sörbom (1996) and comprises twelve observed variables grouped into three latent factors: self-esteem, depression, and impulsiveness, with a total of 204 samples. Our algorithm learns the correct causal structure (including the measurement model and the structure model) for both datasets by first identifying the dimension of latent support as two in the political efficacy dataset and four in the depress dataset.

For the political efficacy data Aish and Jöreskog (1990), by identifying the support of latent variable to be two, one can identify the causal cluster {'NOCARE', 'TOUCH', 'INTEREST'}, {'NOSAY', 'VOTING', 'COMPLEX'}. These clusters correspond to the latent variables 'EFFICACY' and 'RESPONSE', respectively. In our implementation, we set the significance level to 0.0015. The result is aligned with the ground truth provided in Jöreskog and Sörbom (1996).

For the depress dataset, the ground truth structure Jöreskog and Sörbom (1996) includes three latent factors: Self-esteem, Depression, and Impulsiveness, with the corresponding observed clusters:

- {'SELF1', 'SELF2', 'SELF3', 'SELF4', 'SELF5'};
- {'DEPRES1', 'DEPRES2', 'DEPRES3', 'DEPRES4'};
- {'IMPULS1', 'IMPULS2', 'IMPULS3'}.

In our implementation, we utilize a bootstrapping resampling approach to enhance the statistical properties of the data. Following the extended results outlined in Appendix E, we first identify the dimension of support for the factors Self-esteem and Depression as four, and set the dimension of support for Impulsiveness at three. The significance level is set to 0.025. The results of our algorithm are presented as follows.

- {'SELF1', 'SELF2', 'SELF3', 'SELF5'};
- {'DEPRES1', 'DEPRES3', 'DEPRES4'};
- {'IMPULS1', 'IMPULS2', 'IMPULS3'}.

One can see that our algorithm can learn three causal clusters corresponding to three latent factors. Such a result shows that our method finds all latent variables from the depress data. In the latent structure learning process, the PC-TENSOR-RANK algorithm outputs the fully connected graph of the three latent factors, indicating the absence of conditional independence (CI) relations between them. One possible reason is there may be more potential factor interaction structure Salles et al. (2024).

# J   Broader Impacts

The proposed *Tensor Rank Condition* identifies the causal structure of latent variables in the discrete LSM model. Our theoretical results extend the identification bounds of causal discovery with latent variables and expand the application scope of latent variable models. For instance, in psychology and social sciences, our algorithm can learn the causal structure of the latent factors of interest from

observed data (e.g., survey data), enabling researchers to better understand the causal mechanisms behind them. Additionally, based on the discovered causal relationships, one can design more effective intervention strategies to improve mental health or stimulate the market economy.

## K Limitation

The proposed method can hardly be applied to the high-dimensional discrete data. Therefore, how to relax this restriction and make it scalable to high-dimensional real-world datasets would be a meaningful future direction.

