# OpenReview forum: "Learning Discrete Latent Variable Structures with Tensor Rank Conditions"
_NeurIPS.cc/2024/Conference — NeurIPS 2024 poster_

### Official Review · Reviewer_Upr4 · 2024-07-09

**Soundness:** 3
**Presentation:** 2
**Contribution:** 3
**Rating:** 6
**Confidence:** 2

**Summary:**

This study develops new methods for identifying causal structures in discrete latent variable models using rank constrain. By leveraging this nontrivial algebraic property, the authors propose criteria and algorithms for discovering hidden causal relationships. They validate their approach through simulations.

**Strengths:**

1. As far as I know, the tensor rank condition is a novel contribution. Unlike previous on rank deficiency, this method can handle non linear relations.
2. The experiments are done on both simulated and real-world dataset.

**Weaknesses:**

I wonder if some assumptions are a bit strong such as the faithfulness assumption and the three pure child assumption and I wonder how they play into practical scenarios.

**Questions:**

1. How does this paper compare to works like [1] where rank constraint is also used in a non-linear setting?

[1] Kong, Lingjing, et al. "Learning Discrete Concepts in Latent Hierarchical Models." *arXiv preprint arXiv:2406.00519* (2024).

**Limitations:**

See weakness.

---

> ### Author Rebuttal · Authors · 2024-08-06
>
> We appreciate your careful review and suggestions and would like to thank you for your positive assessment of our work.
>
> >Q1: I wonder if some assumptions are a bit strong such as the faithfulness assumption and the three pure child assumption and I wonder how they play into practical scenarios.
>
> **A1**: We would like to mention that to render the identifiability of the latent variable structure, we introduce the assumptions of the three pure children and faithfulness.
>
> Regarding the three-pure child assumption, it is a commonly used condition, well-studied in both linear and discrete models, as demonstrated by Silva et al., 2006, Kummerfeld and Ramsey, 2016, and Chen et al., 2024 (more related works can be found in the introduction of our manuscript). It is worth noting that the pure child assumption is often valid when data is collected through questionnaires, which is a common practice in fields such as social science, psychology, and medical science. For a more detailed discussion of the three-pure children assumption, please refer to the general response.
>
> Regarding the faithfulness assumption, it plays a significant role in constraint-based methods, ensuring that the learned structure accurately represents a causal graph. It is considered a standard assumption, which is known to hold in simple systems such as linear Gaussian models [Meek (1995)]. This concept has been extensively discussed in the literature [Spirtes et al., 2000; Spirtes and Glymour, 1991].
>
> We appreciate your comments and will discuss these assumptions further in the revision.
>
> >Q2:
> How does this paper compare to works like [1] where rank constraint is also used in a non-linear setting?
>
> **A2**: We would like to clarify that there are two main differences compared to [1].
>
> First, the tensor rank condition in our work can capture more d-separation information among observed variable sets. For example, the graphical implications of tensor rank allow us to infer the d-separation among any pair of variables within the variable set $\mathbf{X}_p$, whereas the rank of the probability table only infers the d-separation between two variables (or vectors). In particular, when only two-way tensors are considered, [1] can be seen as a specific case of our approach.
>
> Second, in [1], the observed variables are continuous, making the recovery of the distribution of discrete variables untestable. Additionally, the model in [1] does not allow for 'triangle structures' among the latent variables. In contrast, our work does not require the recovery of the distribution of discrete variables and allows for any dependence among latent variables. Furthermore, [1] assumes that the cardinality of latent support is the same, whereas our work allows for different cardinalities of latent support (see Appendix F).
>
>
> Reference:
>
> [1] Kong, Lingjing, et al. "Learning Discrete Concepts in Latent Hierarchical Models." arXiv preprint arXiv:2406.00519 (2024).

---

> > ### Comment · Reviewer_Upr4 · 2024-08-12
> >
> > Thanks for your rebuttal. I will keep my score leaning towards acceptance.

---

> > > ### Author Response · Authors · 2024-08-12
> > > **Thank you for your response**
> > >
> > > Dear Reviewer Upr4,
> > >
> > > Thank you for your positive comment. We believe our paper is improved as a direct result of your comments.
> > >
> > > Sincerely,
> > >
> > > Authors

---

### Official Review · Reviewer_Snze · 2024-07-10

**Soundness:** 3
**Presentation:** 2
**Contribution:** 3
**Rating:** 6
**Confidence:** 3

**Summary:**

This paper aims to learn latent causal models with discrete random variables. To this end, a tensor rank condition on contingency tables of observed variables is used. Specifically, the paper establishes connections between the d-separation of the observed variables and the “tensor rank” of the said random variables. Subsequently, this property is leveraged to construct an algorithm that learns (i) the measurement model, that is the connections from the latent variables to observed variables, and (ii) the structure model (the edges between the latent variables) up to Markov equivalence class. The main structural assumptions for the results are purity (no edges between the observed variables) and three observed children per latent node, and other standard causal assumptions, e.g., faithfulness, are adopted in the paper.

**Strengths:**

The literature mostly focuses on learning linear latent models. Instead, this paper aims to address discrete latent models for which the existing work is scarce (the paper outlines that they are limited to very restrictive structures, e.g., trees).

The whole paper builds on making a nice observation: the tensor rank of the contingency tables of the joint dist. of the observed variables are connected to the support of the variables that d-separate them. This observation is substantiated in the main result Theorem 3.3. All the subsequent results (mainly the algorithms) build on this main observation, which leads to rather simple proofs. I find this simplicity as a strength. The practical implementation of the proposed method is also clearly explained using existing techniques to test the tensor rank.

**Weaknesses:**

- *Structural limitations are quite restrictive*. Specifically, the purity assumption that there are no edges between the observed variables is a serious limitation. On one hand, this is understandable as the existing works usually have similar or stronger restrictions. On the other hand, having partial results for the cases where this assumption is violated would have made the paper significantly stronger. I see that the paper touches on this via an example and leaves it as future work, nevertheless, it’s a big limitation of the current paper.
- On a related note, the assumptions are not convincingly justified. For instance, three-pure children assumption is just taken for granted without proper discussion (I acknowledge the reference to similar work, please see the questions section).
- This is relatively a minor concern: I enjoyed the flow of the paper, but the writing could be improved for sure. I elaborate on this in questions.

**Questions:**

**Critical points on assumptions**
- *Allowing edges between observed variables*: It seems like extending the results to impure structures would require a different approach. Because the propositions -- the backbone of finding the measurement model -- are very specific to the purity assumption. In the discussion section, an example is provided and the extension is left for future work. I think you can be more open about it, e.g., what are the challenges, what are the missing pieces? For instance, in the example in Section 7, are you also able to determine the lack of edges between any $(X_i,X_j)$ pair except $(X_4,X_5)$?
- *Structural constraints in the literature*: I think you need to be more direct and provide exact requirements/structural constraints and results. For instance, if I recall correctly, Cai et al. and Xie et al. require 2 pure children per latent whereas this paper requires 3 pure children (I am aware that those papers are for linear latent models). Can you elaborate on the similarities and differences between the roles of these assumptions? On key difference from related work, L89 says that the are no constraints on the latent structure. Is this difference w.r.t. binary latent variable case (Gu and Dunson (2023) and Gu (2022)) or all the previous work including the linear latent variable models?
- In Appendix D, the full rank assumption (Assumption 2.4) discussion is informative and reasonable. However, a similar discussion for the other assumptions is missing. For instance, in the example in Section 7, the possibility of working on an impure structure is explained but failure cases (and why they happen) are not explained. I think it’s equally important to explain those cases. On a related note, the role of 3 pure children assumption is not discussed in the paper (please point out if I missed it).
- *Support of latent variables*: The paper says that *”For simplicity, we focus on the case where all latent variables have the same number of categories, with extensions provided in Appendix F”*. Some clarification can be helpful. I skimmed through Appendix F, and it seems like the extension to “latent variables with different numbers of categories” is immediate without extra assumptions. If this is indeed the case, then I suggest the authors explicitly state it in the main paper. If the extensions require more assumptions and/or more sophisticated techniques, again, be more clear about it so the reader can understand if it’s a limitation or not.


**Experiments related**:

- I see that the definitions of metrics are in Appendix I (which should be referred to in L288 to help the reader). It would have been nicer to squeeze them into the main paper.
- Adding a small section for real data applications (Section 6) without sufficient details does not add much value. I think it can be safely moved to the appendix, where the details are given, which would also open space for a more self-contained simulation section.
-  I think your method should be able to handle an arbitrary number of children as long as there are at least 3. So, you could have tested different sizes for different latent variables instead of a fixed number of 3 or 4. For instance, each latent variable can have at least 3, at most 6 observed variables, which could have allowed you to evaluate whether there is a difference between the performance in learning measurement and structure models.
- For such sparse and small graphs, 50k samples are a bit too aggressive. Perhaps starting from 1k would show the limits of the approach and help us to understand the performance better. For instance, at “mismeasurements” metrics, even 5k sample seem to work well, it would have been interesting to see what is the limit.

**Relatively minor notes**:
- *Definition 2.1 (Discrete LSM)*: Is this a proper name? I don’t like calling the specified model “discrete LSM” simply because it is too general. Assumptions are clearly stated in the definition, but calling the model just discrete LSM, especially under the purity and three-pure child assumptions is not great,  especially when we think about the potential future work and positioning of this paper in the literature.
- *Theorem 4.7*: Please be careful when using $p$ in notations, $\mathbf{X}_p$ and $\mathbf{L}_p$ notations are a bit confusing. At first reading, $p$ reads as a set, e.g., in Example 4.8, $\mathbf{X}_p$ implies $p=\\{1,10,4,5,7,8\\}$, but $\mathbf{L}_p=\\{L_2,L_3\\}$, so it is not a set notation.
- *Conclusion Section 8*: Overall, I liked this conclusion. That being said, being more direct and recalling the main structural assumptions would have been good. For instance, the limitations paragraph in Appendix K is very clean — I appreciate it. Since it’s just 2.5 lines, I’d suggest to include in the main paper

**Typos etc.**:
- Prop 4.3: I think Rule 2 statement is missing “if”. Also, why $L_i$ instead of $L$?
- Theorem 4.7: varaible
- Theorem 4.9: TESNOR
- L177: missing period
- L800: TENSKR
- CP decomposition: please spell out at the first appearance
- Figure 8: make the markers larger, please.

**Limitations:**

Limitations are mostly addressed. I suggest the authors to see my comments on questions to further clarify them.

---

> ### Author Rebuttal · Authors · 2024-08-05
>
> Thank you for your insightful and valuable questions and for spending the time and effort on this review. We will respond to these issues point by point.
>
> >Q1: Allowing edges between observed variables: ... I think you can be more open about it, e.g., what are the challenges, what are the missing pieces?
>
> **A1**: One of the challenges in impure structure is accurately identifying the number of latent variables. As shown in Figure 1 in our attached PDF, the rank constraints over the probability tensor can be equal by the graphical implication of tensor rank, hindering the identification of latent variables. Although we can detect the absence of edges between pairs of variables (as in the structure described in Sec. 7), a careful search strategy is still required, along with proper assumptions on the cardinality of latent support, to distinguish these equivalent classes. We will address this issue in our future work.
>
> >Q2: Structural constraints in the literature: I think you need to be more direct and provide exact requirements/structural constraints and results. ... Can you elaborate on the similarities and differences between the roles of these assumptions?
>
> **A2**: In the linear non-Gaussian model, Cai et al. and Xie et al. show that the identification of the measurement model only relies on the two-pure children assumption, due to the availability of higher-order statistics from non-Gaussianity. However, in the linear Gaussian model, the three-pure children assumption is necessary, due to the indistinguishable equivalence classes in two-pure condition by Tetrad constraints (as noted by Silva et al. and Kummerfeld et al.).  Interestingly, when only four-way tensors are considered in discrete models, the tensor rank condition serves as a 'nonlinear' version of the Tetrad constraints, resulting in similar structural limitations. Moreover, for discrete latent variable models, Gu and Dunson (2023) and Gu (2022) address only partial structures and their identifiability. For instance, they focus on either measurement models (under the same three-pure children assumption) or pyramid structures, without providing a comprehensive approach for all possible structures. We will include this discussion in the `Related Works' Section.
>
> >Q3: In Appendix D, ... a similar discussion for the other assumptions is missing. For instance, in the example in Section 7, the possibility of working on an impure structure is explained but failure cases (and why they happen) are not explained.
>
> **A3**: The three-pure children assumption is a sufficient condition that ensures the identifiability of latent variables using the tensor rank condition over a four-way tensor (see more details in the general response). A similar reasoning applies in linear Gaussian latent variable models (e.g., Silva et al. and Kummerfeld et al.).
>
> In the presence of an impure structure, the latent variable may not be correctly identified (see **A1**). Additionally, impure structures can arise when the collected variables have direct causal relationships. For example, in a questionnaire survey, the observed variables ‘occupations' and ‘income' may exhibit an impure structure because the attributes of different occupations can directly impact income.
>
> >Q4: Support of latent variables: … all latent variables have the same number of categories, with extensions provided in Appendix F”. ... If this is indeed the case, then I suggest the authors explicitly state it in the main paper.
>
> **Q4**: Thank you for your careful observation. We will add the following illustrations to Section 4 to avoid confusion.
>
> "For simplicity, we focus on the case where all latent variables have the same number of categories. The result can be directly extended to cases with different numbers of categories by sequentially testing the cardinality of the latent support (see details in Appendix F)."
>
> >Q5&6: I see that the definitions of metrics are in Appendix I. It would have been nicer to squeeze them into the main paper.
>
> **A5&6**: Thank you for your suggestion. We will move the real-world application section into the appendix and add the details of evaluation metrics into the simulation experiment section.
>
> >Q7&8: I think your method should be able to handle an arbitrary number of children as long as there are at least 3. …. Perhaps starting from 1k would show the limits of the approach and help us to understand the performance better.
>
> **A7&8**: Due to limited space, please see our response in the `Supplementary Experiments' part of the general response section.
>
> >Q9: Definition 2.1 (Discrete LSM): Is this a proper name?
>
> **A9**: Thank you for your constructive suggestion. We will revise "discrete LSM" to "three-pure discrete LSM (abbreviated as *discrete 3PLSM*) in the revision."
>
> >Q10: Theorem 4.7: Please be careful when using $p$ in notations.
>
>  **A10**: We have revised them to $L_q$ and $X_p$ to avoid confusion.
>
> >Q11: Conclusion Section 8: Overall, I liked this conclusion. That being said, being more direct and recalling the main structural assumptions would have been good.
>
> **A11**: We have added a recall of the three-pure children assumption and revised the last paragraph in the conclusion as follows:
>
> "However, the proposed method can hardly be applied to high-dimensional discrete data and only applies to pure-children structures. Therefore, relaxing these restrictions and making them scalable to high-dimensional real-world datasets and more general structural constraints, such as impure structures and hierarchical structures, would be a meaningful future direction."
>
> >Q12: Typos etc.
>
> **A12**: Thank you very much for your careful review. We have corrected these issues in the revision.
>
> We sincerely thank the reviewer for their careful review and thoughtful suggestions, which have been very instructive. We will incorporate these discussions in the revision to improve our manuscript. If any explanations remain unclear, we welcome further discussion.

---

> > ### Comment · Reviewer_Snze · 2024-08-09
> >
> > I thank the authors for the well-written rebuttal, it addressed my questions and confusions clearly. I increased my score accordingly to “weak accept”.
> > This rebuttal deserves an explanation for why I am not giving a higher score: It's mainly because I think the considered setting is still somewhat limited — even if the assumptions are justified within the context of the paper’s results.

---

> > > ### Author Response · Authors · 2024-08-09
> > > **Thanks for you kind response**
> > >
> > > Dear Reviewer Snze,
> > >
> > > We are glad that most of your concerns have been addressed, and we are really grateful that you raised your score to “weak accept”.
> > >
> > > Sincerely,
> > >
> > > Authors

---

### Official Review · Reviewer_hs7U · 2024-07-12

**Soundness:** 3
**Presentation:** 3
**Contribution:** 3
**Rating:** 8
**Confidence:** 4

**Summary:**

This paper studies the problem of learning causal structures among latent variables from discrete observational data. The author presents a tool, termed the tensor rank condition, to establish the connection between rank constraints of the probability tensor and d-separation relations in the causal graph. The proposed tool appears simple and effective. Based on the tensor rank condition, the author proposes a two-stage algorithm that first learns the causal clusters to identify latent variables and then tests the conditional relations among latent variables using their measured variables. The proposed algorithm extends the identification bound of discrete latent variable models, and the experimental results demonstrate the efficiency of the proposed methods.

**Strengths:**

1. The paper is clearly written and well-organized.

2. The proposed tensor rank condition establishes the connection between algebraic constraints and d-separation relations in discrete causal models. This tool has the potential to explore more causal discovery problems in discrete causal models

3. Compared to traditional methods of the linear latent variable model, such as rank constraints of the covariance matrix, the tensor rank condition takes a meaningful step in latent variable modeling in more general cases.

4. The proposed algorithm looks simple but effective, addressing the identification of discrete latent variable models.

**Weaknesses:**

1. For the sufficient observation assumption, it seems that the cardinality of the observed variable support can be equal to the cardinality of the latent support, as discussed in Remark F.4. Could you clarify this?

2. Why is the Three-Pure Child Variable Assumption required? In my opinion, the tensor rank condition can test the d-separation relations between only two observed variables, as shown in Figure 1. Thus, one can learn the causal cluster from a two-pure child variable assumption. Please correct me if I am wrong. If this assumption does not hold, what happens to the output of your algorithm?

3. In Proposition F.3, $r$ is easily confused with the concept of the cardinality of the support of a single latent variable. It is recommended to change the expression like $\tilde{r}$.

**Questions:**

typo: In Figure 2(b), LVM should be LSM.

**Limitations:**

see weakness

---

> ### Author Rebuttal · Authors · 2024-08-06
>
> We appreciate your valuable comments and suggestions and thank you for your positive assessment of our work.
>
> >Q1: For the sufficient observation assumption, it seems that the cardinality of the observed variable support can be equal to the cardinality of the latent support, as discussed in Remark F.4. Could you clarify this?
>
> **A1**: Thank you for your suggestion. You are correct. We only require the cardinality of the observed variables to equal that of the latent variables. We will revise this in the revision.
>
> >Q2: Why is the Three-Pure Child Variable Assumption required? In my opinion, the tensor rank condition can test the d-separation relations between only two observed variables, as shown in Figure 1. Thus, one can learn the causal cluster from a two-pure child variable assumption. Please correct me if I am wrong. If this assumption does not hold, what happens to the output of your algorithm?
>
> **A2**: Thanks for raising these critical questions. If there are only two pure-measured children for each latent variable, it can result in an indistinguishable structure. For instance, as shown in the proof of Proposition 4.3, the tensor rank condition for a three-variable probability tensor leads to equivalence classes (Fig. 4 (a) $\sim$ (c)). However, with the four-way tensor, the cluster cannot be identified due to the structure illustrated in Fig. 5. Thus, the three-pure measured variable assumption is a sufficient condition to ensure the identification of the measurement model. If this assumption is not met, our algorithm may output a causal structure where the number of latent variables is smaller than the true number. However, to test the conditional independence (CI) relations among the latent variables, only two pure children for each latent variable are necessary (See Theorem 4.7). Thus, given the causal cluster with only two pure children for each latent variable, the CI relations can be tested, and the structure model remains identifiable. One can see more discussions in the general response.
>
> > Q3: In Proposition F.3,  is easily confused with the concept of the cardinality of the support of a single latent variable. It is recommended to change the expression like r.
>
> **A3**: Thank you for your suggestion. We have revised it in the revision.
>
> > Q4: typo: In Figure 2(b), LVM should be LSM.
>
> **A4**: Thank you for your suggestion. We have corrected it in the revision.

---

> > ### Comment · Reviewer_hs7U · 2024-08-12
> >
> > Thanks for the detailed response. My questions have been well addressed. As far as I know, the tensor rank condition is the first to establish the connection between algebraic constraints and d-separation in discrete causal models. Based on this, the identification algorithm for discrete LSM is both simple and effective. I believe these results are meaningful to the NeurIPS community. I raise my score accordingly.

---

> > > ### Author Response · Authors · 2024-08-12
> > > **Thank you very much for your positive comment**
> > >
> > > Dear Reviewer hs7U,
> > >
> > > Thank you very much for your positive comment! We believe our paper is improved as a direct result of your comments.
> > >
> > > Sincerely,
> > >
> > > Authors

---

### Official Review · Reviewer_tR91 · 2024-07-13

**Soundness:** 3
**Presentation:** 3
**Contribution:** 2
**Rating:** 4
**Confidence:** 4

**Summary:**

This paper studies the problem of learning latent variable structure in the discrete LSM measurement model. My understanding is that the paper operates under the following assumptions:
1) All latent variables are discrete
2) There are no connections between observed variables (i.e. observed variables are independent when conditioned on latent variables
3) In the unknown causal structure graph, each observed variable has exactly one latent variable parent.
4) The joint probability distribution of observed variables is fully known, and there is no noise in the measurement.
5) Each latent variable has at least three pure variables as children.
6) Other technical assumptions.

The authors show that under such assumptions the unknown causal structure can be recovered from an algorithm that studies tensor rank conditions on contingency tables for an observed variable set.

**Strengths:**

The problem studied in this paper covers an important case in causal structure learning literature. The measurement model with discrete latent variables is an important special case of causal structure models observed in practice, hence identifiability results and structure learning algorithms are important for this problem.

The paper is well-written, and the claims are sound. Theoretical results are supported by experiments.

**Weaknesses:**

I am a bit concerned about the novelty of the proposed results and methodology. In particular, [1] seems to obtain a stronger result under similar, if not weaker, assumptions, and this paper does not cite [1].

[1] proposes a structure learning algorithm for measurement model with discrete latent variables, under what seems to be a weaker set of assumptions. In particular [1] does not require each latent variable to have at least three pure children and does not require that each observed variable has only one latent parent. Those assumptions seem quite strong.

The technique used in [1] to identify causal structure is also similar in flavor, and also relies on studying the rank of joint probability tensor (through numbers k(S)).

Considering that [1] operates under a significantly more general setup while relying on similar techniques and ideas, could the authors please clarify what is the main novelty of this paper compared to [1]?

[1] Learning latent causal graphs via mixture oracles
B Kivva, G Rajendran, P Ravikumar, B Aragam - Advances in Neural Information Processing Systems, 2021

**Questions:**

see above

-----
During rebuttal score was increased 3 --> 4

---

> ### Author Rebuttal · Authors · 2024-08-06
>
> Thanks for your careful and valuable comments. We will respond to these issues point by point.
>
> > Q1: [1] proposes a structure learning algorithm for measurement model with discrete latent variables, under what seems to be a weaker set of assumptions. In particular [1] does not require each latent variable to have at least three pure children and does not require that each observed variable has only one latent parent. Those assumptions seem quite strong.
> In particular, [1] seems to obtain a stronger result under similar, if not weaker, assumptions, and this paper does not cite [1].
>
>
> **A1**: Thank you for pointing us to this interesting work. We would like to discuss its connection with our work.
>
> The main difference is that the identifiability of [1] requires to assume that the "Mixture Oracle" is known (i.e., the mixture model over $\mathbf{X}$ is identifiable), without discussing how to obtain it, especially in discrete data while we do not require such an assumption. We would like to argue that the requirement of the Mixture Oracle may lead to a weaker identifiability result than ours due to the strict conditions in obtaining the Mixture Oracle. For instance, [2] indicates that a sufficient and necessary condition for identifying the parameters of a discrete mixture model is the presence of $2K-1$ strongly separate variables, where $K$ is the number of mixture components. This structural condition is more stringent than compared to our requirements.
> The reason is that identifying the mixture model requires learning the latent distribution $P(\mathbf{L})$, while our method does not rely on this.
> Furthermore, as discussed in Sec. 6 in [1], learning mixture models is a nontrivial problem. The author in [1] uses the approximate method, such as the K-means algorithm, to estimate the number of mixture components $k(S)$ for all $S \subset X$, which can affect the accuracy of the estimates in practice. Meanwhile, the K-means algorithm cannot be directly used in the discrete data, we thus do not include it in our baselines.
>
>
> >Q2:  The technique used in [1] to identify causal structure is also similar in flavor, and also relies on studying the rank of joint probability tensor (through numbers k(S)).
> Considering that [1] operates under a significantly more general setup while relying on similar techniques and ideas, could the authors please clarify what is the main novelty of this paper compared to [1]?
>
> **A2**: We would like to clarify the following points:
>
> **Identification based on Tensor Decomposition.**
>  Roughly speaking, [1] depends on unique tensor decomposition (actually, a technology of parameter estimation) and requires partial information from a Mixture Oracle. However, it is difficult to obtain the Mixture Oracle and unbiased parameter estimates. In contrast, our theoretical approach focuses on tensor rank constraints across different sets of variables, without directly relying on unique tensor decomposition or any prior information about the Mixture Oracle.
>
> Specifically, in [1], the identification of the measurement model relies on unique tensor decomposition, such as the generalized Kruskal's theorem. To construct such a tensor, we require partial information from a Mixture Oracle, such as $k(S)$, where $k(S) = dim(Pa(S))$ according to observation 2.7 in [1]. It is important to note that $k(S)$ for any subset $S \subset X$  cannot be estimated from the tensor rank. As demonstrated by the graphical implications of tensor rank, the tensor rank is determined by the cardinality of the conditional set. That is, $Rank(P(S)) \neq dim(Pa(S)) = k(S)$. For example, consider a structure $L_1 \to L_2$, $X_1$ is children of $L_1$ and $\{X_2, X_3\}$ are children set of $L_2$. In this case, the tensor rank of the joint probability tensor over $S = \{X_1, X_2, X_3\}$ is $|supp(L_2)|$. According to the mixture oracle in [1], it requires that $k(S) = |supp(L_1, L_2)|$. In contrast, our theoretical result does not directly rely on the unique decomposition of tensors, but rather examines tensor rank constraints over various variable sets.
> A key challenge with the decomposition-based method is the difficulty in obtaining unbiased parameter estimates. Our method avoids this issue by focusing on the number of parameters, specifically the cardinality of latent support, making it a more practical solution.
>
> **Novelty.**
> In our paper, we propose a statistically testable method, termed the tensor rank condition, which establishes the connection between tensor rank and d-separation among observed variables. The tensor rank condition is applicable to general discrete causal models and is not restricted to measurement models. This represents one of our main contributions compared to [1]. Based on the tensor rank condition, we provide an identification algorithm that can determine the causal structure of discrete latent structure models with a theoretical guarantee. Unlike [1], our algorithm offers a solution for discrete latent variable models without requiring any prior knowledge about the mixture model, thereby elegantly eliminating the need for a mixture oracle. Moreover, our method offers a theoretical guarantee and is simple and efficient due to the testability of the tensor rank condition.
>
> **Acknowledgements.**
> Thank you for raising the constructive questions, which have inspired us to further explore discrete latent variable models. We will incorporate these discussions in the revised version, specifically in the 'Related Works' section. If you believe your concerns have been addressed, we kindly request that you consider adjusting your initial score accordingly. If any explanations remain unclear, we are open to further discussion.
>
> Reference:
>
> [1]. Kivva B, Rajendran G, Ravikumar P, et al. Learning latent causal graphs via mixture oracles.
>
> [2]. Tahmasebi B, Motahari S A, Maddah-Ali M A. On the identifiability of finite mixtures of finite product measures.

---

> > ### Comment · Reviewer_tR91 · 2024-08-12
> >
> > I am grateful to the authors for their detailed and constructive rebuttal. I am still concerned about the novelty of the results of this paper as compared to [1]. Let me try to provide some more details regarding my concern.
> >
> > The Mixture Oracle in [1] is essentially only used to count the number of components $k(S)$ in the "mixture decomposition" in
> >
> > (*) $P(S) = \sum_{\ell\in supp(pa(S))}P(S|pa(S) = \ell)P(pa(S) = \ell)$,
> >
> > or in other words to estimate the size of the $supp(Pa(S))$ if assumption 2.4.a from [1] is added, where S is a subset of observed variables. See eq. (2-3) in [1] and Remark 2.6.
> >
> > > We would like to argue that the requirement of the Mixture Oracle may lead to a weaker identifiability result than ours due to the strict conditions in obtaining the Mixture Oracle
> >
> > The Mixture Oracle in the setup of this paper can be easily obtained from Assumption 2.4 (in this paper) and the assumption about three pure children made in this paper. To see this,
> > 1) observe that $P(S|L = \ell)$ is a rank-1 tensor of the form $\sum_{\ell \in supp(pa(S))}\bigotimes_{X_i \in S}P(X_i|pa(X_i) = \ell_j)$
> > 2) Let $X_i, X_j, X_k$ be pure children of vertex $L_t$, then by assumption 2.4 columns $P(X_i|pa(X_i))$ can be recovered as first modes of minimum rank tensor decomposition of $P(X_i, X_j, X_k)$.
> > 3) Observe that if vectors in each of sets A and B are linearly independent, then all vectors $a\otimes b$ for $a\in A, b\in B$ are linearly independent. Applying this recursively, we get that all vectors $\bigotimes_{X_i \in S}P(X_i|pa(X_i) = \ell_j)$ are linearly independent, and we know the full set of those vectors from 2). Therefore, we can find unique decomposition (*) and count the number of components with non-zero coefficients (in practice, coefficients less than some threshold).
> > 4) note that the decomposition in 3) is not a minimum rank decomposition, as was also pointed out by the authors in the rebuttal, so Theorem 3.3 cannot be used to estimate the number of components.
> >
> > I also suspect that the desired mixture oracle should exist under significantly weaker assumptions than what I outlined above; in particular, using three pure children seems to be an overkill, but I have not spent time trying to simplify these assumptions.
> >
> > ----
> >
> > Based on the above, Theorem 3.3 appears to be the key novel result of this paper. The algorithm in the sections that follow operates under significantly stronger assumptions than [1]. Theorem 3.3 is an exciting result and, indeed, may provide a more robust approach than what [1] offers. That said, I still think that the assumptions used in this paper are somewhat strong compared to other results in the literature, as also pointed out by other reviewers.
> >
> >  After the author's rebuttal, I will increase my current score to 4.

---

> ### Author Response · Authors · 2024-08-12
> **Thank you very much for your response**
>
> Thank you very much for your response. Please allow us to take a little time to clarify further.
>
> We want to argue that $supp(Pa(S))$ cannot be directly estimated even under Assumption 2.4 and the three-pure children assumption when the latent structure is unknown and the subset $S$ has different latent variables.
>
> We agree that in your example, when we *known* the three observed $X_i$, $X_j$ and $X_k$ is the pure children of $L$, the $supp(Pa(S))$ can be recovered through the tensor decomposition. However, when $S$ has different latent variables and the *unknown* causal structure, the minimum rank tensor decomposition does not necessary correspond to the $supp(Pa(S))$ and it will produce multiple equivalence class as we discussed in the proof of Prop. 4.3 in our paper, such that $supp(Pa(S))$ can not be identified.
>
>
>
> Consider the structure in Fig. 4(c) in our paper, without loss generality, let $L_2 \to L_1, L_2 \to L_3$, $L_1 \to X_j, L_2 \to X_i$ and $L_3 \to X_k$. Let $S = \{X_i, X_j, X_k\}$. As such, we will have the following decomposition:
>
>
> $$\begin{align}P(S) =& \sum_{i = 1} ^{|supp(L_1)|} \left( \sum_{L_2} P(X_j, L_2|L_1=i) \right) \otimes P(X_i|L_1=i) \otimes \left( \sum_{L_3} P(X_k, L_3|L_1=i) \right) \cdot P(L_1=i)\\\\
> =& \sum_{i = 1} ^{|supp(L_1)|} \left( \sum_{L_2} P(X_j, L_2|L_1=i) \right) \otimes P(X_i|Pa(X_i)=i) \otimes \left( \sum_{L_3} P(X_k, L_3|L_1=i) \right) \cdot P(L_1=i)
> \end{align}$$
> where the vectors $\sum_{L_2} P(X_j, L_2|L_1=i)$ and $\sum_{L_3} P(X_k, L_3|L_1=i)$ also are linear independent. In this case, we can find a unique decomposition (based on Kruskal's theorem), but not follow the form of (*), i.e., $P(S) \ne \sum_{\ell \in supp(Pa(S))} P(S|Pa(S) = \ell) P(Pa(S) = \ell)$.
>
>
> Moreover, in our work, the conditional independence between latent variables can be tested by the tensor rank condition (Theorem 4.7), without assuming the existence of the mixture model and recovering the distribution of latent variables.
>
> We are very grateful for your patient response. Please feel free to reach out if you have any further questions.

---

> ### Comment · Reviewer_tR91 · 2024-08-12
>
> I believe authors misunderstood points 3 and 4 in my comment above. I do not claim that (*) is obtained via minimal rank decomposition, as authors aim to refute in their response. 3 and 4 above use that any vector has a unique decomposition in a basis in a linear space.
>
> To give a simplified example: Let $a_1, a_2 \in R^n$ be linearly independent and $b_1, b_2 \in R^m$ be linearly independent, then vectors $a_1\otimes b_1, a_2\otimes b_1, a_1\otimes b_2, a_2\otimes b_2$ are all linearly independent (as vectors in $R^{nm}$) and for a vector $T = \alpha_1 a_1\otimes b_1 +\alpha_2 a_2\otimes b_2 +\alpha_3 a_3\otimes b_3$ one can uniquely recover this decomposition if $a_i$ and $b_i$ are known, even if it tensor has smaller rank (say, 2).
>
> One can recover vectors $P(X_i|pa(X_i))$ by looking at minimum rank decompositions of triples of variables and use it to recover necessary decompositions of $P(S|Pa(S))$ (and hence get mixture oracle), as I outlined above.
>
> Please feel free to reach out if something is still not clear.

---

> > ### Author Response · Authors · 2024-08-12
> > **Thanks for the clarification in helping us algining this issues**
> >
> > Thanks for the clarification in helping us algining this issues.
> > It seems that you treat the $Pa(X_i|Pa(X_i))$ as the basis, thereby the tensor $P(S|Pa(S))$ can be recovered.
> >
> > However, we want to argue that without knowing the structure information we will have different basis when we probe at different triples and we do not know which one is the "correct" basis (here $Pa(X_i|Pa(X_i))$) we are looking for.
> >
> > For example, when there exists a minimum rank decomposition for $X_i, X_j, X_k$, the basic can be equal to $Pa(X_j|L_i) \neq Pa(X_j|Pa(X_j))$ (see the example in the previous response). In this case, we do not know if $Pa(X_j|L_i)$ is $Pa(X_j|Pa(X_j))$.
> >
> >
> > Moreover, we want to show that, without the structure information, $Pa(X_i|Pa(X_i))$ may not be identifiable due to violation of the unique decomposition condition.
> >
> > For example, consider a structure with three observed variables $X_i, X_j, X_k$ and $L_1, L_2, L_3$ are their latent parent, respectively. To obtain the minimum rank decompositions of $X_i, X_j, X_k$ for estimating $Pa(X_i|Pa(X_i))$, it must satisfy the Kruskal's condition, that is, $Rk(P(X_i|L))+Rk(P(X_j|L))+Rk(P(X_k|L)) >= 2r+2$, where $r$ is the cardinality of latent support L, where $L = \[{Pa(X_i), Pa(X_j), Pa(X_k)\}]$. However, when the cardinality of the latent and observed support both are $d$, one can see that $Rk(P(X_i|L))+Rk(P(X_j|L))+Rk(P(X_k|L)) = 3d < 2d^{3}+2$. Therefore,  $Pa(X_i|Pa(X_i))$ cannot be recovered due to a violation of the Kruskal condition. And it also violate the condition of 2K-1 strongly separate variables in [2], a sufficient and necessary condition for identifying the parameters of a discrete mixture model.
> >
> >
> > We are very grateful for your patient response. Please feel free to reach out if you have any further questions.

---

> ### Comment · Reviewer_tR91 · 2024-08-13
>
> I am grateful for your response.
>
> > However, we want to argue that without knowing the structure information we will have different basis when we probe at different triples and we do not know which one is the "correct".
>
> It is actually easy to identify the "correct" one. Let's fix $X_i$ and look at triples with different $X_j, X_k$. If minimum rank decomposition has more than d components we can immediately refute triple as incorrect, which handles your concern about uniqueness of the decomposition. By Kruskal's theorem, those triples that admit decomposition of rank at most d will have a unique decomposition. Decompose all such tensors and compile a list of first-mode components for each of them.
> Note that $P(X_i|L = \ell) = \sum_{t} P(X_i|L = \ell, Pa(X_i) = t) P(Pa(X_i) = t| L = \ell) =  \sum_{t} P(X_i| Pa(X_i) = t) P(Pa(X_i) = t| L = \ell)$ and all $P(Pa(X_i) = t| L = \ell)$ are non-negative. This means that each vector of the form $P(X_i|L = \ell)$ belongs to the convex hull of the "correct" set of vectors $P(X_i| Pa(X_i) = t)$. Since all P(X_i| Pa(X_i) = t) are linearly independent by assumption 2.4, they are the unique "defining directions" of the convex cone formed by all first mode components of tensor decompositions of $P(X_i, X_j, X_k)$ with a fixed $i$. Note also that there is no ambiguity about the direction (sign) of the vectors in the decomposition, as we expect every vector participating in the decomposition to have positive entries.
> Note that no structure information is used, and one do not need to assume that all latent variables have the same support, etc.
>
> > Moreover, we want to show that, without the structure information, $Pa(X_i|Pa(X_i))$ may not be identifiable due to violation of the unique decomposition condition.
>
> By assumptions of this paper, the triples of variables $X_i, X_j, X_k$ which are of interest to construct "mixture oracle" have unique decomposition of rank at most $d$ ($d$ here can be replaced with the size of the support of observed variables, so knowledge of $d$ is not needed). All other triples can be ignored.
>
> Please let me know if you have any questions.

---

> ### Author Response · Authors · 2024-08-13
> **Many thanks for your question**
>
> **Dear Reviewer tR91,**
>
> Thank you for your effort and your time. We would like to argue that by fixing $X_i$ and looking at triples with different $X_j, X_k$, we may not be able to figure out that the triple is incorrect.
>
> For example, consider a simple structure $L_1 \to L_2 \to L_3$, where each latent varaible has three pure children, corresponding to $X_{123}$, $X_{456}$, and $X_{789}$, respectively. When we fix $X_1$ and look at any $X_j, X_k \in \overline{X_1}$, one can see that there is still a minimum rank decomposition with d components (where d represents the cardinality of latent support). This is because any three observed variables can be d-separated by only one latent variable in this structure (see the proof of Prop. 4.3).  For instance, let $X_j=X_4$ and $X_k = X_7$, which is the example in the previous response. Thus, we cannot check if the basic $P(X_j|L) $ is actually $P(X_j|Pa(X_j))$ from the decomposition.
>
>
> Please let us take this opportunity to highlight our contributions, in order to situate this work in the rich literature:
>
> 1. We introduce a statistically testable tool, termed the  *tensor rank condition* , which establishes a connection between rank constraints and d-separation relations within a discrete model.
> 2. Utilizing the tensor rank condition, we achieve the identification of the measurement model under the three pure children assumption.
> 3. Furthermore, based on the tensor rank condition, we complete the identification of the structural model, requiring only two pure children for each latent variable.
>
> In [1], the authors leverage information from a mixture oracle and perform tensor decomposition to identify the measurement model, recovering the distribution to learn the latent variable structure. However, the identification approach in this work relies on the existence of a mixture oracle, and estimating the mixture oracle is a non-trivial task.
>
> Most importantly, we want to emphasize that the tensor rank condition is not restricted to any specific mixture model and does not rely on information from such models. The discrete causal structure can be discovered using the tensor rank condition, which allows for identifying edges between observed variables and even permits causal directions from observed to latent variables. We believe that the tensor rank condition opens up new research directions and offers promising avenues for developing search algorithms for latent variable researchers.
>
>
> Moreover, as we discussed in the general response, the three pure children assumption is merely a sufficient condition for identifying the measurement model by tensor rank. We also explored potential extensions to more general conditions, such as impure structures or hierarchical structures, for identifiability. We discussed that the tensor rank condition has the capacity to handle these cases, which we outline as directions for future work.
>
>
> If you have further feedback, we hope to read it and hope for the opportunity to respond to it.  We highly appreciate your engagement in the discussion.

---

> > ### Comment · Reviewer_tR91 · 2024-08-13
> >
> > >  We would like to argue that by fixing X_i and looking at triples with different X_j, X_k, we may not be able to figure out that the triple is incorrect.
> >
> > I explained above how this can be done by taking convex hull of vectors in the first mode of the decomposition. I think the authors in the subsequent paragraph refute something I did not claim.
> >
> > >However, the identification approach in this work relies on the existence of a mixture oracle, and estimating the mixture oracle is a non-trivial task.
> >
> > Performing the steps I described above provides the desired mixture oracle and gives identifiability results out of the box.
> >
> > > We introduce a statistically testable tool, termed the tensor rank condition , which establishes a connection between rank constraints and d-separation relations within a discrete model.
> >
> > *I agree that this is a novel and an interesting contribution of this paper!*
> >
> > It is nice to see how this can be used for the identification of the structure. However, as I outlined above, the identifiability results for the measurement model essentially follow from weaker assumptions from prior work. It is possible that the proposed method is more statistically robust under the stronger assumptions of this paper, but this is not how this paper describes its contribution.

---

> > > ### Author Response · Authors · 2024-08-13
> > >
> > > We are very grateful for your thoughtful comments, which we basically agree with.  Below please let us share some thoughts.
> > >
> > > In contribution [1], the authors show that to learn the measurement model, only the $k(S)$ in the mixture oracle is required.
> > >
> > > In our previous response, we mentioned that recovering the complete information about the mixture oracle generally requires strict assumptions--we hope you also agree with it, since it involves identifying more parameters of the mixture oracle.
> > >
> > > Regarding your point about the identification of the measurement model under a weaker assumption, we acknowledge that it is indeed possible since only $k(S)$ is needed. We appreciate this insight. At the same time, we would like to highlight that our method offers a more statistically robust approach for achieving this under the three pure children assumption, which you also pointed out. This assumption often holds in practice, especially when data is gathered through questionnaires, which is common in fields including social science, psychology, and healthcare.
> > >
> > > That being said, we feel that our work goes beyond just this aspect. As you mentioned, we provide a testable tool that offers a stronger ability to discover causal structures. Additionally, it's nice to note that, given the measurement model, the causal structure among latent variables is identifiable with only two pure children needed for each latent variable. We believe this is another major contribution of our work.
> > >
> > > We are very grateful for your patient response. Please let us know if you have any further questions--we greatly appreciate the opportunity to discuss with you.

---

### Author Rebuttal · Authors · 2024-08-06

**General Response**

We thank the reviewers for their efforts in reviewing our manuscript and for the insightful comments and suggestions. Please see below for our general response.

**Sufficient Condition: Three-Pure Children Assumption.**
To ensure the identifiability of latent variables, the pure children assumption is generally required and well-studied in the related literature (e.g., Silva et al., 2006; Kummerfeld and Ramsey, 2016; Chen et al., 2024; Bollen, 2002; Bartholomew et al., 2011), as discussed in the first paragraph of the introduction in our manuscript. Besides, the three-pure children assumption is necessary to ensure that the latent variable can be identified by the tensor rank condition. The reason is as follows. If there are only two pure-measured children for each latent variable, it can lead to an indistinguishable structure. For example, as shown in the proof of Proposition 4.3, the tensor rank condition for a three-variable probability tensor has equivalence classes (Fig. 4 (a) $\sim$ (c)). However, for the four-way tensor, the cluster cannot be identified due to the structure shown in Fig. 5. Thus, the three-pure measured variable assumption is a sufficient condition to ensure the identification of the measurement model. If this assumption is violated, our algorithm may output a causal structure where the number of latent variables is smaller than the true number. However, to test the conditional independence (CI) relations among the latent variables, only two pure children for each latent variable are required (See Theorem 4.7). That is, given the causal cluster with only two pure children for each latent variable, the CI relations can still be tested, and the structure model remains identifiable.

**Supplementary Experiments (Response to #Reviewer Snze).**
Due to limited space, we are responding to the supplementary experimental part here. We aim to (i) evaluate the performance of our algorithm with different numbers of measured variables, and (ii) explore the behavior of our algorithm when the number of samples is much smaller. The results are reported in Tables 1 to 4 in our attached PDF.

In Table 1 $\sim$ 2, one can see that our method still achieves good performance even when the number of measured variables is different. In fact, it can enhance the accuracy in determining the number of latent variables, as a latent variable can be identified from only its three pure children using the tensor rank condition.


Moreover, in Table 3 $\sim$ 4, the performance of our method is not good when the sample size is small (e.g., 1k sample size) because the tensor rank is inaccurately calculated in such cases. The 'mismeasurements' metrics are also lower because most of the observed variables are grouped into the same cluster in the procedure of finding causal clusters. As the sample size increases, our method achieves better performance.

---

### Decision · Program_Chairs · 2024-09-25

**Decision:**

Accept (poster)

**Comment:**

This paper studies learning latent variable structures under discrete data. This problem is well-studied in the literature, with numerous identifiability result. During the discussion, it was pointed out by Reviewer tR91 that many results in the submission follow from previous work by Kivva et al that is not cited and which is less restrictive. This negates some claim by the authors, for example, that "few identifiability results exist and are mostly only applicable in strict case".

There was an extensive back and forth. In the end, reviewers agreed that in spite of the misleading presentation of prior work, this paper indeed introduces some new ideas, namely, a statistically testable tool that connect rank constraints and d-separation. To quote the discussion:

"It is possible that the proposed method is more statistically robust under the stronger assumptions of this paper, but this is not how this paper describes its contribution."

I agree with this point, and the paper could be rejected on this. **I am willing to recommend accept if the authors (significantly) revise the paper to properly situate their contributions in light of this (and other, see [1-3] below) prior work and also clarify their contributions according to the discussion with Reviewer tR91.**

[1] Causal Discovery under Latent Class Confounding https://arxiv.org/abs/2311.07454

[2] Identifiability of Product of Experts Models https://arxiv.org/abs/2310.09397

[3] Causal Inference Despite Limited Global Confounding via Mixture Models https://arxiv.org/abs/2112.11602